# Sex difference in parental risk of suicide attempt during and after pregnancy in Sweden

Yihui Yang [1] ✉, Emma Bränn [1,2], Emma Fransson [3,4], Krisztina D. László[5,6], Fang Fang[1], Fotios C. Papadopoulos[7], Unnur A. Valdimarsdóttir [1,8,9], Alkistis Skalkidou[3] & Donghao Lu [1] ✉

Whether the risks of maternal and paternal suicide attempt during and after pregnancy differ remains unclear. Here, in this nationwide register-based study in Sweden (2,196,276 pregnancies), we defined the year before conception, pregnancy and the year after birth and estimated week-specific incidence rate ratios (IRRs). We identified 7,469 (1.39 per 1,000 person-years) suicide attempts among mothers and 8,338 (1.62 per 1,000 person-years) among fathers. Compared with the corresponding week in the preconception period, mothers had a lower risk of suicide attempt during and after pregnancy (with the lowest IRR of 0.14 (0.11–0.17) at first week postpartum); fathers' risk of suicide attempt remained largely stable before childbirth, but a lower risk was observed during the first 10 postpartum weeks (IRRs ranging from 0.69 (0.58–0.81) to 0.91 (0.84–0.99)), followed by a higher risk in the later postpartum period (IRRs ranging from 1.10 (1.01–1.21) to 1.72 (1.33–2.24)). Compared with fathers, mothers had a lower risk of suicide attempt during and after pregnancy (for example, IRR of 0.22 (0.18–0.28) at first week postpartum). Compared with the general population, the sex difference of suicide attempt is reversed during and after pregnancy, suggesting pregnancy or childbirth may have a more pronounced association with suicide attempt among mothers than fathers.

Suicide is a public health concern that substantially affects families and societies[1]. Reducing suicide rates has been set out as a target by both the United Nation Sustainable Development Goals and the World Health Organization Global Mental Health Action Plan[2]. In most parts of the world, completed suicide is more common among men than women[3,4], while for suicide attempts, the gender difference is reversed[4]. A better understanding of the sex/gender differences in suicide attempt, the strongest risk factor of completed

suicide, may be informative to tailor gender-based strategies for suicide prevention.

For both women and men, transition into parenthood is a life-changing period that may substantially affect their mental health. Parental suicide attempt during the perinatal period is fortunately rare[5,6], but is associated with severe health outcomes for the affected individual and their offspring[7–9]. Compared with fathers, mothers experience more physiological changes during the perinatal period[10,11] and

[1]Institute of Environmental Medicine, Karolinska Institutet, Stockholm, Sweden. [2]Center for Epidemiology and Community Medicine, Region Stockholm, Stockholm, Sweden. [3]Department of Women's and Children's Health, Uppsala University, Uppsala, Sweden. [4]Department of Microbiology, Tumor and Cell Biology, Karolinska Institutet, Stockholm, Sweden. [5]Department of Global Public Health, Karolinska Institutet, Stockholm, Sweden. [6]Department of Public Health and Caring Sciences, Uppsala University, Uppsala, Sweden. [7]Department of Medical Sciences, Psychiatry, Uppsala University, Uppsala, Sweden. [8]Center of Public Health Sciences, Faculty of Medicine, University of Iceland, Reykjavík, Iceland. [9]Department of Epidemiology, Harvard T.H. Chan School of Public Health, Boston, MA, USA. ✉e-mail: yihui.yang.2@ki.se; donghao.lu@ki.se

are more likely to be affected by perinatal depression[12]. However, mothers may also gain from the clinical monitoring in maternal care and the activated social support during pregnancy and after childbirth[11,13]. This may counterbalance vulnerabilities to suicide attempt, and even reverse the female excess in suicide attempt, during this period.

However, the knowledge on sex differences in parental suicide attempt during pregnancy or after childbirth is limited. Several studies have reported that childbirth was associated with a lower risk of suicide attempt and completed suicide among mothers[14–18], although this association was not observed among those with severe psychiatric disorders[19]. Only one study estimated the prevalence of paternal suicide ideation and attempt in the antepartum and postpartum periods[6], without comparing with a reference period. To our knowledge, no study has investigated how transition into parenthood may reshape sex differences in suicide attempt. To this end, we described and compared rates of maternal and paternal suicide attempts before, during and after pregnancy and investigated whether mothers and fathers differ in risk of suicide attempt in these time periods. While we recognize the diversity in contemporary families, we focused on heterosexual couples in the present study.

## Results

We included 2,196,276 pregnancies from 1,236,816 mothers and 1,175,674 fathers (Supplementary Fig. 1). In the study sample, mothers were younger and had higher educational attainment but lower income than the fathers ($P < 0.001$; Table 1). In addition, mothers were more likely to have a history of depression and other psychiatric disorders or a history of suicide attempt ($P < 0.001$). During the follow-up of over 10 million person-years, we identified 7,469 (incidence rate (IR) of 1.39 per 1,000 person-years) suicide attempts among mothers and 8,338 (1.62 per 1,000 person-years) among fathers. Among mothers, IR of suicide attempt was 2.69, 0.74 and 0.85 per 1,000 person-years in the preconception, antepartum and postpartum period, respectively; among fathers, IR of suicide attempt was 1.88, 1.45 and 1.53 per 1,000 person-years in the preconception, antepartum and postpartum periods, respectively (Supplementary Table 1).

### Secular trends in parental suicide attempt

For both mothers and fathers, the annual standardized incidence rate (SIR) of suicide attempts in the preconception and antepartum periods increased from 2001 to 2010 (adjusted $P < 0.05$), and decreased thereafter in the antepartum and postpartum periods (adjusted $P < 0.05$; Fig. 1 and Supplementary Table 2).

### Sex differences in parental suicide attempt

Among mothers, the SIR by week was largely stable in the preconception period, slightly decreased during pregnancy and slightly increased throughout the postpartum period (adjusted $P < 0.05$; Fig. 2 and Supplementary Table 3). Among fathers, the SIR was largely stable over the three periods (adjusted $P < 0.05$; Fig. 2 and Supplementary Table 3).

Compared with the preconception period, there was a decreased risk of suicide attempts throughout the antepartum and postpartum periods among the mothers, with the lowest risk observed at first week postpartum, with an incidence rate ratio (IRR) of 0.14 (95% confidence interval (CI) 0.11–0.17) and adjusted $P < 0.001$ (Fig. 3 and Supplementary Table 4; parameter results are provided in the Supplementary Methods). Compared with corresponding weeks in the preconception period, the risk of paternal suicide attempt was similar during the antepartum period, significantly decreased during the first 10 weeks postpartum and increased in the later postpartum period (Fig. 3; statistics are provided in Supplementary Table 4).

Compared with expectant fathers, expectant mothers had a similar risk of suicide attempt throughout the preconception period after adjusting for factors potentially associated with suicide attempt—such as a history of psychiatric disorders and prior suicide attempts—whereas a decreased risk was observed during the antepartum and postpartum periods, with the lowest IRR noted during the first week postpartum (IRR 0.22, 95% CI 0.18–0.28, adjusted $P < 0.001$) (Fig. 4 and Supplementary Table 5; parameter results are provided in the Supplementary Methods). A similar pattern was seen for IRD; however, the largest IRD was seen at the 16th to 21st week antepartum.

### Secondary analyses

An increasing trend of postpartum suicide attempt was indicated among adolescent fathers (aged 11–19 years) (IRR 1.06 (1.00–1.11), $P = 0.037$), but the association is no longer significant after adjusting for multiple comparison (adjusted $P = 0.096$). Meanwhile, IR among adolescent mothers was stable (IRR 1.00 (0.96–1.03), $P = 0.891$, adjusted $P = 0.940$) (Supplementary Fig. 2). In addition, we found a decreasing trend of suicide attempts from 2006 onwards among both fathers and mothers with and without depressive disorders (Supplementary Fig. 3 and Supplementary Table 6). However, this decreasing trend was driven mainly by individuals with a history of suicide attempt (Supplementary Fig. 4 and Supplementary Table 7). In addition, the pattern of weekly SIR was similar to main analysis when stratified by depressive disorders (Supplementary Fig. 5 and Supplementary Table 8) and history of suicide attempt (Supplementary Fig. 6 and Supplementary Table 9). The IRRs comparing antepartum and postpartum periods to preconception period were similar regardless of depressive disorders (Supplementary Fig. 7 and Supplementary Table 10) or history of suicide attempt (Supplementary Fig. 8 and Supplementary Table 11).

Overall, we identified 7,178 (45%) cases of suicide attempt by poisoning, 2,239 (14%) by cutting or piercing, and 330 (2%) by falling (Supplementary Table 12). Compared with fathers, the risk of suicide attempt by poisoning, cutting or piercing among mothers was significantly lower in most weeks in the antepartum and postpartum periods; there is a similar pattern in suicide attempt by falling, but only significant during the middle weeks of the antepartum and postpartum periods. Notably, a significantly higher risk of suicide attempt by poisoning was observed among mothers in the preconception period (Supplementary Fig. 9 and Supplementary Table 13). IRR calculated by comparing antepartum and postpartum periods to average IR of preconception period yielded largely similar results on trends, whereas the paternal IRR was no longer statistically significant (Supplementary Fig. 10 and Supplementary Table 14). Furthermore, when excluding week 53 from the analysis, the rise in paternal absolute IR in late postpartum period was less pronounced, yet the CIs overlapped with those from the primary analysis (Supplementary Fig. 11 and Supplementary Table 15) whereas the IRRs changed minimally (Supplementary Figs. 12 and 13 and Supplementary Tables 16 and 17).

## Discussion

In this nationwide cohort including 2,196,276 pregnancies from 1,236,816 mothers and 1,175,674 fathers in Sweden, we found a lower IR of suicide attempt among mothers compared to fathers during and after pregnancy, which is opposite to the sex differences in the general population. Compared with the year before pregnancy, mothers had a decreased risk of suicide attempt throughout pregnancy and one year postpartum. However, fathers had a decreased risk after childbirth, and a higher risk of suicide attempt in the later postpartum period only when compared with corresponding weeks in preconception year.

### Secular trends of suicide attempt

Three studies, all from the USA, reported on the secular trend of perinatal suicide attempt among mothers[20–22]. Two studies reported an increasing rate of perinatal suicide attempt from 2006 to 2018[20,21], whereas one reported a stable trend between 2006 and 2012[22]. We did not identify any studies that reported on the secular trend of paternal suicide attempts. In our study, in some periods, SIR of perinatal suicide attempt increased from 2001 to 2010 but decreased afterwards

**Table 1 | Characteristics at the start of preconception, antepartum and postpartum period between mothers and fathers: a nationwide population-based cohort study in Sweden, 2001–2021**

| | | Mothers | | Fathers | | Chi square | Degree of freedom | P value |
|---|---|---|---|---|---|---|---|---|
| | | PYs | % | PYs | % | | | |
| Total | | 5,394,157 | 100.00 | 5,163,556 | 100.00 | | | |
| Age group (year) | 11–19 | 91,196 | 1.69 | 29,252 | 0.57 | 55,6131.56 | 5 | <0.001 |
| | 20–24 | 677,192 | 12.55 | 326,485 | 6.32 | | | |
| | 25–29 | 1,676,347 | 31.08 | 1,187,179 | 22.99 | | | |
| | 30–34 | 1,845,242 | 34.21 | 1,742,963 | 33.76 | | | |
| | 35–39 | 907,971 | 16.83 | 1,174,251 | 22.74 | | | |
| | ≥40 | 196,209 | 3.64 | 703,426 | 13.62 | | | |
| Calendar year | 2001–2005 | 1,280,222 | 23.73 | 1,252,895 | 24.26 | 740.64 | 3 | <0.001 |
| | 2006–2010 | 1,330,858 | 24.67 | 1,284,403 | 24.87 | | | |
| | 2011–2015 | 1,377,865 | 25.54 | 1,312,123 | 25.41 | | | |
| | 2016–2021 | 1,405,212 | 26.05 | 1,314,135 | 25.45 | | | |
| Educational level (years) | <10 | 562,726 | 10.43 | 599,578 | 11.61 | 118,401.15 | 3 | <0.001 |
| | 10–12 | 2,036,515 | 37.75 | 2,427,582 | 47.01 | | | |
| | ≥13 | 2,656,123 | 49.24 | 2,028,559 | 39.29 | | | |
| | Unknown | 138,793 | 2.57 | 107,837 | 2.09 | | | |
| Country of birth | Sweden | 4,183,632 | 77.56 | 4,058,701 | 78.60 | 2,606.11 | 3 | <0.001 |
| | Europe | 429,991 | 7.97 | 413,737 | 8.01 | | | |
| | Other | 780,142 | 14.46 | 690,733 | 13.38 | | | |
| | Unknown | 392 | 0.01 | 385 | 0.01 | | | |
| Civil status | Cohabitating | 4,425,947 | 82.05 | 4,298,744 | 83.25 | 2,651.59 | 1 | <0.001 |
| | Non-cohabitating | 968,210 | 17.95 | 864,812 | 16.75 | | | |
| Income level[a] | Quantile 1 | 1,197,662 | 22.20 | 874,152 | 16.93 | 90,656.86 | 5 | <0.001 |
| | Quantile 2 | 1,126,081 | 20.88 | 979,029 | 18.96 | | | |
| | Quantile 3 | 1,080,117 | 20.02 | 1,029,836 | 19.94 | | | |
| | Quantile 4 | 1,037,563 | 19.23 | 1,074,490 | 20.81 | | | |
| | Quantile 5 | 932,421 | 17.29 | 1,164,380 | 22.55 | | | |
| | Unknown | 20,313 | 0.38 | 41,669 | 0.81 | | | |
| Primiparity[b] | No | 2,876,277 | 53.32 | 2,759,100 | 53.43 | 13.30 | 1 | <0.001 |
| | Yes | 2,517,880 | 46.68 | 2,404,456 | 46.57 | | | |
| History of psychiatric disorders | No | 4,175,163 | 77.40 | 4,336,495 | 83.98 | 84,583.95 | 2 | <0.001 |
| | Depressive disorders | 439,452 | 8.15 | 237,725 | 4.60 | | | |
| | Other psychiatric disorders | 779,542 | 14.45 | 589,336 | 11.41 | | | |
| History of suicide attempt | No | 5,235,196 | 97.05 | 5,050,495 | 97.81 | 6,027.79 | 1 | <0.001 |
| | Yes | 158,961 | 2.95 | 113,061 | 2.19 | | | |
| Season[c] | Spring | 1,345,424 | 24.94 | 1,288,110 | 24.95 | 1.40 | 3 | 0.705 |
| | Summer | 1,419,427 | 26.31 | 1,360,092 | 26.34 | | | |
| | Fall | 1,363,383 | 25.28 | 1,304,811 | 25.27 | | | |
| | Winter | 1,265,923 | 23.47 | 1,210,543 | 23.44 | | | |

PY, person-year. [a]Quantiles 1, 2, 3, 4 and 5 represent 0–20th percentile (lowest), 21–40th percentile, 41–60th percentile, 61–80th percentile and 81–100th percentile (highest) of the income distribution, respectively. [b]For mothers, we collected data on parity, and for fathers, we collected data on number of pregnancies of the women who share biological children with them. [c]The season of spring includes months from March to May, summer includes June to August, fall includes September to November and winter includes December to February. Two-sided chi-square tests were used to compare characteristics between fathers and mothers.

among both mothers and fathers. In 2010, Sweden launched a national programme for maternal postpartum depression screening[23]. Such screening may reasonably result in early detection of depression and prevention of suicide attempts among postpartum mothers. However, we did not find a drop in SIR specific to the postpartum period among mothers, indicating that other factors are at play. Sweden has also implemented a national programme for suicide prevention since 2008, including distributing knowledge on suicide-reducing strategies[24]. This may contribute to the recent decline in the risk of suicide attempt in the general population in Sweden[25], as well as in parents in

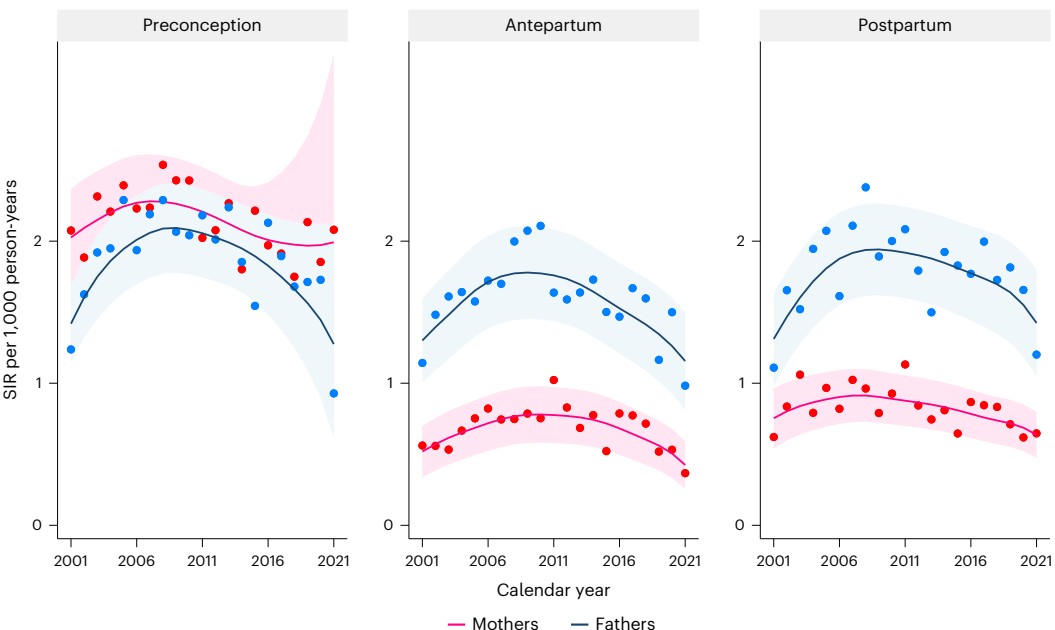

**Fig. 1 | SIR of parental suicide attempt before, during and after pregnancy: a nationwide register-based study in Sweden, 2001–2021.** The incidence rate was standardized by distribution of age group of the accumulated person-years during follow-up. Locally weighted scatterplot smoothing (bandwidth 0.8) was used to estimate the trend (solid line), while the dots indicate the yearly incidence rates. The shaded areas indicate the 95% CI. *P* values for trend test are two-sided and adjusted for multiple comparison (Supplementary Table 2).

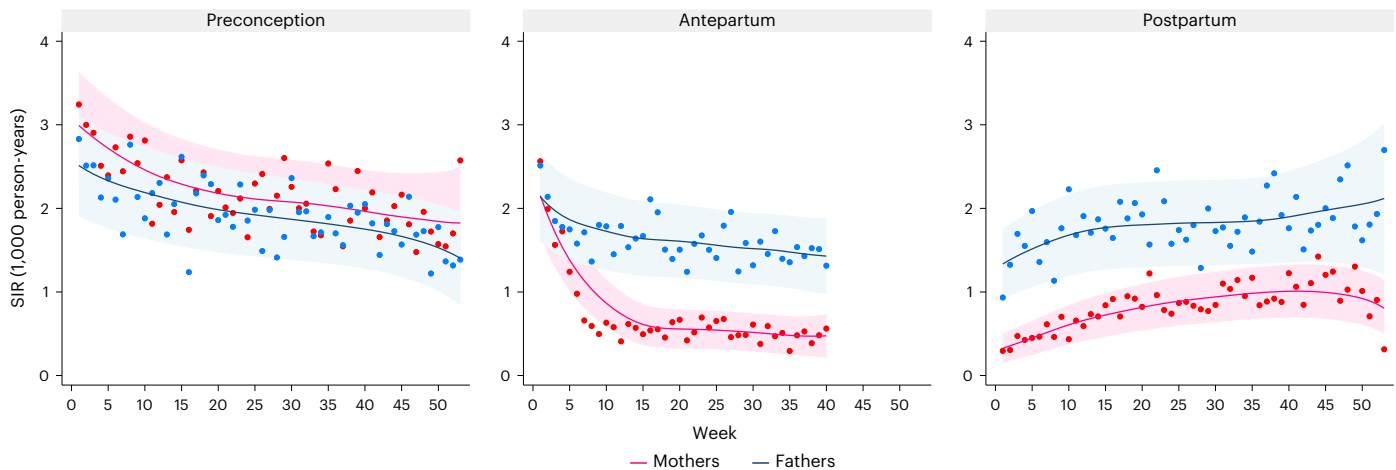

**Fig. 2 | SIR of parental suicide attempt before, during and after pregnancy by week: a nationwide register-based study in Sweden, 2001–2021.** The follow-up during pregnancy was censored at week 40 due to few subsequent cases. Incidence rates were standardized by distribution of age group and calendar period of the accumulated person-days during follow-up. Locally weighted scatterplot smoothing (bandwidth 0.8) was used to estimate the trend (solid line), while the dots indicate the weekly incidence rates. The shaded areas indicate the 95% CI. *P* values for trend test are two-sided and adjusted for multiple comparison (Supplementary Table 3).

the perinatal period. In addition, childbirth rate in Sweden has declined since 2010[26], possibly due to financial uncertainty and labour market insecurity[27]. We hypothesize that individuals who were healthier and economically more advantaged may have been more likely to choose to engage in pregnancy during this period than those less advantaged.

Teenage pregnancy has been linked to several adverse health outcomes among adolescent mothers[28,29]. Studies have reported a decreasing number of children born to teenage mothers[30], but a stable or slight increase in teenage pregnancy rates in Sweden[31,32]. Our results indicated an increasing rate of postpartum suicide attempt over calendar time among adolescent fathers, although the association is no longer significant after adjusting for multiple comparison.

No increasing trend was indicated for adolescent mothers. Further research is needed to focus on adolescent fathers and to explore sex differences in postpartum mental health among teenage parents.

### Sex difference in preconceptional suicide attempt

Preconception is considered as a relative healthy period when expectant mothers plan for pregnancy[33], and possibly for fathers as well. Our study observed a largely stable SIR of suicide attempt over the weeks in preconception period. This suggests such 'health bias' may not notably affect the weekly trend of suicide attempts before pregnancy. In addition, we found a similar risk of suicide attempt among mothers and fathers over weeks throughout the preconception period. The trend of

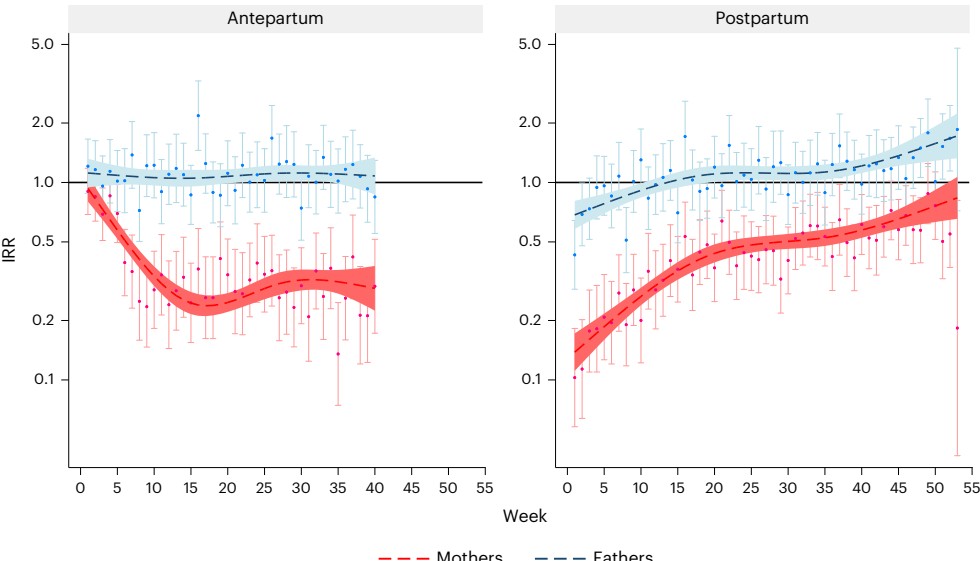

**Fig. 3 | IRR of parental suicide attempt during and after pregnancy compared with the corresponding week before pregnancy: a nationwide register-based study in Sweden, 2001–2021.** The analyses are based on aggregated data from 2,196,276 pregnancies (1,236,816 mothers and 1,175,674 fathers). The follow-up during pregnancy was censored at week 40 due to few subsequent cases. The dots represent the actual IRRs, and the error bars represent their 95% CI. *P* values are two-sided and adjusted for multiple comparison (Supplementary Table 4). The dashed lines represent results when using restricted cubic spline with 4 knots, placed at 5th percentile, 35th percentile, 65th percentile and 95th percentile of distribution of weeks on the relevant *x* axes. The shaded areas indicate 95% CI of the smoothed results. Note that a logarithmic scale was used. The IRRs were adjusted for country of birth, age, calendar year, education level, civil status, category of income, primiparity, history of psychiatric disorders, history of suicide attempt and season, all derived at the start of each period.

weekly SIRs in preconception period among fathers somewhat echoed the trend among mothers. This is consistent with previous literature reporting a positive link on suicide risk between spouses, possibly due to stress brought by suicide of a partner, or shared environmental factors between the cohabitating partners or selective mating[34,35].

## Sex differences in antepartum suicide attempt

Few studies reported a decreased rate of suicide attempt, or completed suicide, among women during pregnancy compared with the general female population[14,15]. However, preexisting risk factors of suicide can also affect the chance of entering into motherhood[36]. Such confounders were not addressed sufficiently in earlier studies. Similarly to a study in Canada[16], our study found a decreased risk of maternal suicide attempt during pregnancy, when comparing the risk of suicide attempt during pregnancy with the period before pregnancy, which serves as a more valid comparison group in terms of controlling for confounding. In addition, our results also indicated a similar risk of paternal suicide attempt during antepartum period when compared with corresponding weeks in the preconception period. Our results suggest that, compared with corresponding weeks in the preconception period, the risk of suicide attempt decreased during pregnancy among mothers[36], but not among fathers. Despite this, maternal perinatal suicide may involve more violent suicide methods, as it may represent a more severe psychopathology[37,38], and parental suicide attempts are associated with mental health problems in their offsprings[7]. Therefore, close surveillance on suicidality among expectant parents remains critical.

Notably, the reduction in risk of suicide attempt during the antepartum period was observed only among mothers. It is plausible that mothers have a better support network than fathers during pregnancy[39]. For example, mothers usually gain support from female relatives, whereas fathers often turn to colleagues at work[40]. Alternatively, mothers have more contacts with health professionals than fathers during pregnancy. Maternal psychiatric symptoms may be detected earlier, demonstrating the potential benefit of maternity care[41]. In addition, progesterone has anxiolytic effects, and studies on suicidality across menstrual cycles have shown that women tend to commit suicide at premenstrual and menstrual phases, when the progesterone level is low[42,43]. It is plausible that the increased level of progesterone during pregnancy protects mothers against suicide. Although more research is needed to understand the male excess in suicide attempt during pregnancy, clinicians should recognize that female excess in suicide attempt in the general population is reversed to male excess during pregnancy.

## Sex difference in postpartum suicide attempt

Throughout the postpartum period, we found a slightly increasing SIR of suicide attempts among mothers. Adjusting to demands of early parenting creates stress for new parents[44], whereas current postpartum care does not seem to provide enough information on early parenthood[45]. However, more research is needed to further understand the rebound of suicide attempt postpartum.

Compared with the preconception period, mothers still had a decreased risk of suicide attempt throughout the postpartum period, whereas fathers had a decreased risk only in the first 10 weeks postpartum. These differences may be related to the fact that mothers are usually the primary caregivers during the first months and may therefore develop closer emotional bonds with their children than fathers, which in turn may buffer against poor mental health in mothers[36]. In addition, in some qualitative studies, fathers claimed that difficulties in balancing needs of family and work were important sources of stress in the year after birth[46]. In addition, fathers may experience hormonal and neurofunctional alterations after childbirth[47], but more research is needed to understand whether these biological changes are associated with paternal suicide attempt postpartum. As mental health screening is not in place for fathers yet, health providers may stay alert of paternal risk during the postpartum period.

## Strengths and limitations

Our study is a nationwide population-based cohort study, with relatively precise definition of the exposure periods based on delivery date and gestational age. The large sample size provided the possibility to study the temporal pattern of suicide attempt among

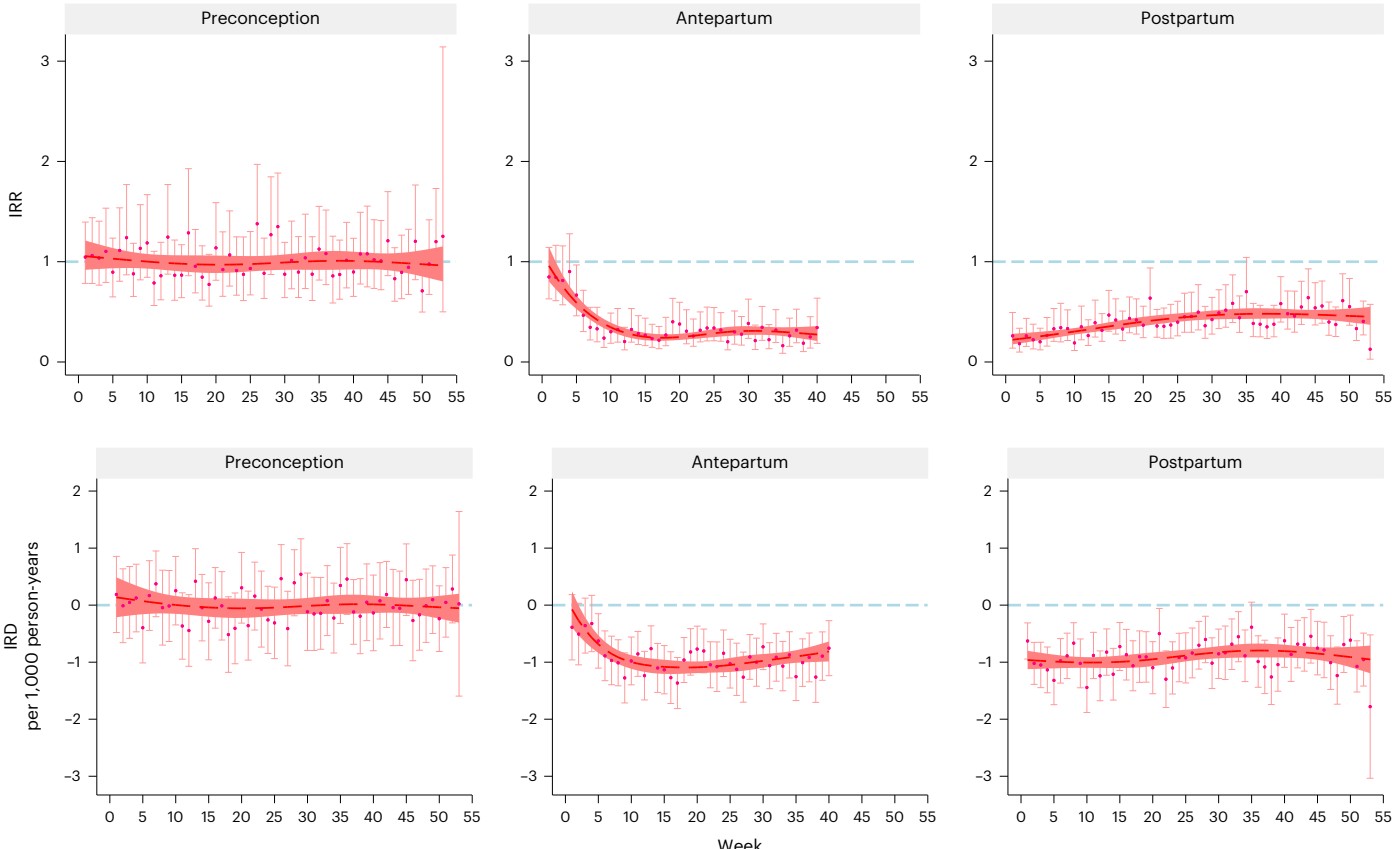

**Fig. 4 | IRR and IRD of suicide attempt among mothers compared with the corresponding week among fathers: a nationwide register-based study in Sweden, 2001–2021.** The analyses are based on aggregated data from 2,196,276 pregnancies (1,236,816 mothers and 1,175,674 fathers). The follow-up during pregnancy was censored at week 40 due to few subsequent cases. The dots represent the actual IRRs and IRDs, and the error bars represent their 95% CI. *P* values are two-sided and adjusted for multiple comparison (Supplementary Table 5). The dashed lines represent results when using restricted cubic spline with 4 knots, placed at 5th percentile, 35th percentile, 65th percentile and 95th percentile of distribution of weeks on the relevant *x* axes. The shaded areas indicate 95% CI of the smoothed results. The IRRs and IRDs were adjusted for country of birth, age, calendar year, education level, civil status, category of income, primiparity, history of psychiatric disorders, history of suicide attempt and season, all derived at the start of each period.

fathers and mothers in a weekly manner, and the comparison with the preconception period allowed us to minimize confounding by parenthood. However, our study has several limitations. First, as the Medical Birth Register (MBR) records only live births and stillbirths that reach 22 weeks of gestation, we cannot assess suicide attempts in pregnancies terminated before 22 weeks of gestation. Second, we may have included some cases of non-suicidal self-harm. However, such events are often milder compared with suicide attempts and are less likely to be recorded in registers. Third, we may have included some cases of fatal suicide if the suicide attempt led to death several days later. In addition, as we can only identify fathers who were alive at least at conception and mothers who were alive by childbirth, we were unable to study completed suicide in the preconception period for both parents or in antepartum period among mothers. Fourth, we had preconception year as the reference period, which may be a mentally and physically healthier period compared with other periods in life[33]. We would thus expect stronger associations for childbirth with suicide attempts if compared with the general population. Sixth, due to data availability, we were unable to identify individuals whose gender does not align with sex assigned at birth. Although our results may not be generalizable to these groups, further research is needed. Seventh, as Sweden is a high-income country with family-friendly policies (for example, free prenatal care, paid parental leave for both parents and other generous welfare benefits during and after pregnancy), our results may generalize only to countries with similar healthcare systems and social welfare.

In conclusion, our results illustrate a lower risk of suicide attempt among both mothers and fathers after childbirth, supporting that childbirth may be a protective factor against suicidality among mothers. Our study also illustrates that the well-established female excess in suicide attempt in the general population is reversed during and after pregnancy. Although perinatal suicidal events are relatively rare, clinical action is needed due to the costly consequence in contrast to the preventable nature. Clinicians may take opportunities of the current scheme of maternal and child health services to identify high-risk individuals, implement suicide risk assessment and enhance suicide prevention strategies for both mothers and fathers.

## Methods

### Study population and design

We conducted a prospective population-based cohort study based on Swedish registers. The Swedish MBR was established in 1973 and includes antenatal and obstetric records for 98% of all births in Sweden[48]. The data from the MBR included year and month, but not date, at delivery due to the data holder's policy. Due to the need of precise timeline for our statistical analyses, we imputed the date at delivery using a validated algorithm based on admission and discharge dates to delivery wards together with parity and mode of delivery (Supplementary Methods). Gestational length was estimated primarily on the basis of the ultrasound assessment performed around the 18th week of gestation[48]. We used delivery date and length of gestation to estimate the start of pregnancy and to define three time windows: 1 year before pregnancy

(preconception period), during pregnancy (antepartum period) and 1 year after delivery (postpartum period). This study is approved by the Swedish Ethical Review Authority (2018/1515-31 and 2021-02775). Informed consent is waived for register-based studies in Sweden.

In Sweden, sex is assigned at birth, although a small number of individuals medically and legally changed their gender later in life[49]. Based on the MBR, we first identified 1,258,824 women and their 2,263,596 pregnancies during 2001–2021 in Sweden. After excluding women who had an invalid personal identification number, missing information on length of gestation, conflicting information (for example, died before start of pregnancy or delivery date) and erroneous records, 1,236,816 women and 2,196,276 pregnancies remained in the analysis (Supplementary Fig. 1). The Multi-Generation Register (MGR) provides information on familial linkages for individuals born since 1932 in Sweden[50] and was used to identify fathers who share biological children with these women. Because fathers were not reported for 2% of children in the MGR, we complemented the identification of fathers as partners of women who reported as cohabitating with 'the-father-to-be' at their first antenatal visit, based on the MBR. Among the 1,213,034 fathers identified, we excluded those who had invalid personal identification numbers or conflicting information, leaving 1,175,674 fathers in the analysis (Supplementary Fig. 1).

Participants were followed from 1 year before pregnancy, moving to Sweden, or 1 January 2001, whichever came later, until 1 year after delivery, death, moving out of Sweden, or 31 December 2021, whichever came earlier.

## Measures

**Ascertainment of suicide attempt.** We identified diagnoses of suicide attempt from the National Patient Register (NPR), both inpatient and outpatient register, using the International Classification of Diseases (ICD) codes. Consistent with previous studies[51,52], we included both definite (ICD8/9: E950-E959; ICD10: X60-X84) and indeterminate (ICD8/9: E980-E989; ICD10: Y10-Y34) suicide attempts (Supplementary Table 18). The first suicide attempt during each period was considered as the outcome. Any suicide attempt before the start of each period was used as a covariate. The positive predictive value for injuries in the NPR is 95% (ref. 53), but no validation study has been specifically conducted for suicide attempt in Swedish registers.

**Ascertainment of covariates.** Covariates were derived for mothers and fathers respectively, at the start of each period. Demographic factors, including birth year and country of birth, were collected from the Total Population Register (TPR). Proxies of socioeconomic status, including education level and income, were obtained from the Longitudinal Integration Database for Health Insurance and Labor Market Studies (LISA). Income was further classified into five quantiles, with quantile 1 representing the lowest, that is the 0–20th percentile of the income distribution. Information on civil status was collected from the MBR and the TPR. From MBR, for mothers we collected data on parity, and for fathers we collected data on parity of the woman they shared a biological child with. The national guideline in Sweden recommends that (semi-)structured diagnostic interviews be carried out when diagnosing psychiatric disorders[54]. We identified history of any psychiatric disorder using ICD-8/9/10 codes from NPR; to supplement with psychiatric diagnoses made in primary care by general practitioners, we also retrieved information on filled psychotropic prescriptions using Anatomical Therapeutic Chemical codes based on the Prescribed Drug Register (Supplementary Table 18). Previous studies suggest that the validity of the diagnoses of psychiatric disorders in NPR and primary care is good[55–59]. As Sweden launched in 2010 a national screening programme for maternal perinatal depression, which may influence sex differences in suicide attempt among individuals with depressive disorders, we also categorized psychiatric disorders into depressive disorders and other psychiatric disorders.

## Statistical analyses

Basic characteristics between mothers and fathers were compared using the $\chi^2$ test. In the analysis of the secular trend in risk of suicide attempt, we first calculated annual sex-specific IRs of suicide attempt during preconception, antepartum and postpartum periods, as the number of first suicide attempt divided by accumulated person-years. Then, we calculated SIRs through direct standardization, using the distribution of age group of the accumulated person-years during follow-up between 2001 and 2021 as the standard. The 95% CI for SIR was calculated on the basis of the standard error[60] and two-sided $Z$ value (SIR ± 1.96 × standard error). Locally weighted scatterplot smoothing, with 0.8 as the bandwidth, was used to smooth the SIR and its 95% CIs[61]. As Sweden launched a national screening programme for maternal depression in 2010, for each period and sex, we examined the association between calendar year and IR of suicide attempt before/ at and after 2010 separately and adjusted for groups of age. As a total of 12 comparisons were performed, we adjusted $P$ values using the Benjamini–Hochberg method, to account for multiple testing[62].

As pregnancy length is measured by week in clinical practice, in the analysis of sex differences in suicide attempt, we first calculated sex-specific IRs of suicide attempt by week in the preconception, antepartum and postpartum year. We then estimated SIR through direct standardization, using the distribution of age group and calendar period of the accumulated person-days during follow-up between 2001 and 2021 as the standard. The 95% CI for SIR was calculated on the basis of the standard error[60] and two-sided $Z$ value (SIR ± 1.96 × standard error). Locally weighted scatterplot smoothing, with 0.8 as the bandwidth, was used to smooth the SIR and its 95% CI. In addition, for each period and sex, we performed Poisson regression to examine the association between week and IR of suicide attempts, when adjusting for group of age and calendar year. We applied the Benjamini–Hochberg method to adjust $P$ values over six comparisons.

To compare the week-specific IRs of suicide attempt in the antepartum and postpartum periods with corresponding weeks in the preconception period, we used multivariable Poisson regression to estimate the IRR and 95% CI of maternal and paternal suicide attempt. Comparing weekly rates allows us to control for season, which is a potential confounder to the studied association[63]. IRRs were adjusted for country of birth, age, calendar year, education level, civil status, category of income, primiparity, history of psychiatric disorders, history of suicide attempt and season, all derived at the start of each period. To investigate a potential nonlinear relationship between IRRs and week, we applied restricted cubic spline on week, and placed 4 knots at the 5th percentile, 35th percentile, 65th percentile and 95th percentile of distribution of weeks on the relevant $x$ axes[64], to visualize weekly IRRs. The parameterization can be found in the Supplementary Methods. The 95% CI was calculated on the basis of the standard error estimated from the delta method and $z$ critical value. Due to small number of events beyond week 40 in the antepartum period, we did not test IRRs over week 40 in the antepartum period; therefore, the Benjamini–Hochberg method was applied to adjust for multiple comparisons over a total of 186 tests (40 in the antepartum period, and 53 in the postpartum period for each sex).

Similarly, to illustrate sex differences in the risk of suicide attempts during each period, we used multivariable Poisson regression to estimate the IRR of suicide attempts, comparing mothers with fathers on a weekly basis from preconception through the postpartum period. To shed light on absolute sex differences, we further used Poisson regression to estimate the weekly incidence rate difference (IRD)[65]. To visualize IRRs and IRDs, we also applied restricted cubic spline on week, and placed 4 knots at the 5th percentile, 35th percentile, 65th percentile and 95th percentile of distribution of weeks on the relevant $x$ axes. The parameterization can be found in the Supplementary Methods. The 95% CI was calculated on the basis of the standard error estimated from the delta method and $z$ value. The models were adjusted for country

of birth, age, calendar year, education level, civil status, category of income, parity, history of psychiatric disorders, history of suicide attempt and season, all derived at the start of each period. For both IRR and IRD, the Benjamini–Hochberg method was applied to adjust for multiple comparisons over a total of 146 tests (53 weeks in the preconception and postpartum periods, respectively, and 40 weeks in the antepartum period).

**Secondary analyses.** We estimated the trend in parental suicide attempt by age group. We also performed Poisson regression to test the secular trend. In addition, to illustrate potential risk modification by depressive disorders and history of suicide attempt, we stratified analyses on time-dependent depressive disorders as well as history of suicide attempt. Furthermore, to shed light on potential prevention strategies, we also identified common methods for suicide attempts and calculated method-specific IRRs comparing mothers with fathers by the three most common suicide methods. In addition, because the SIR in the preconception period fluctuated over weeks, we also estimated IRR comparing the antepartum and the postpartum to the preconception period, using the average IR in the preconception year as the reference. Finally, due to the small number of person-years, we excluded week 53 from the analyses, calculated the SIR by week, and estimated IRRs comparing antepartum and postpartum periods with preconception, as well as comparing mothers with fathers during preconception and postpartum. In all secondary analyses, we applied the Benjamini–Hochberg method to adjust for multiple comparisons.

Data were cleaned using SAS, version 9.4 (SAS Institute) and analysed using Stata 18.0 (STATA). To account for the false discovery rate, we applied the Benjamini–Hochberg method to adjust $P$ values. A two-sided adjusted $P < 0.05$ was considered statistically significant.

### Reporting summary
Further information on research design is available in the Nature Portfolio Reporting Summary linked to this article.

## Data availability
The Public Access to Information and Secrecy Act in Sweden prohibits individual-level data being publicly available. Researchers who are interested in replicating this study can apply for individual-level data from TPR, MGR and LISA Sweden (https://www.scb.se/en/services/ordering-data-and-statistics/ordering-microdata/). Data on patient health from the Swedish MBR, NPR and Prescribed Drug Register can be requested through Socialstyrelsen (https://www.socialstyrelsen.se/en/statistics-and-data/registers/).

## Code availability
Analysis coding is available via GitHub at https://github.com/yihuiyang2/ParentalSuicide.

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

## Acknowledgements

This work is supported by the scholarship from the China Scholarship Council (no. 202106100010 to Y.Y.), grant 2020-01003 from the Swedish Research Council (Vetenskapsrådet) (to D.L.), grants 2020-00971 and 2023-00399 from the Swedish Research Council for Health, Working Life, and Welfare (FORTE) (to D.L.) and a grant from Karolinska Institutet Strategic Research Area in Epidemiology and

Biostatistics (to D.L.). The funders had no role in the study design, data collection and analysis, decision to publish or preparation of the manuscript.

## Author contributions

Y.Y. and D.L. conceived of the study and analysed the data. Y.Y. drafted the paper. All authors interpreted the results, reviewed the paper and approved the decision to submit the paper.

## Funding

## Competing interests

The authors declare no competing interests.

## Additional information

**Correspondence and requests for materials** should be addressed to Yihui Yang or Donghao Lu.

# Reporting Summary

## Statistics

For all statistical analyses, confirm that the following items are present in the figure legend, table legend, main text, or Methods section.

| n/a | Confirmed | |
|---|---|---|
| ☐ | ☒ | The exact sample size (*n*) for each experimental group/condition, given as a discrete number and unit of measurement |
| ☐ | ☒ | A statement on whether measurements were taken from distinct samples or whether the same sample was measured repeatedly |
| ☐ | ☒ | The statistical test(s) used AND whether they are one- or two-sided<br>*Only common tests should be described solely by name; describe more complex techniques in the Methods section.* |
| ☐ | ☒ | A description of all covariates tested |
| ☒ | ☐ | A description of any assumptions or corrections, such as tests of normality and adjustment for multiple comparisons |
| ☐ | ☒ | A full description of the statistical parameters including central tendency (e.g. means) or other basic estimates (e.g. regression coefficient) AND variation (e.g. standard deviation) or associated estimates of uncertainty (e.g. confidence intervals) |
| ☐ | ☒ | For null hypothesis testing, the test statistic (e.g. *F*, *t*, *r*) with confidence intervals, effect sizes, degrees of freedom and *P* value noted<br>*Give P values as exact values whenever suitable.* |
| ☒ | ☐ | For Bayesian analysis, information on the choice of priors and Markov chain Monte Carlo settings |
| ☒ | ☐ | For hierarchical and complex designs, identification of the appropriate level for tests and full reporting of outcomes |
| ☒ | ☐ | Estimates of effect sizes (e.g. Cohen's *d*, Pearson's *r*), indicating how they were calculated |

*Our web collection on statistics for biologists contains articles on many of the points above.*

## Software and code

Policy information about availability of computer code

| | |
|---|---|
| Data collection | No software was used in data collection. |
| Data analysis | Data were cleaned using SAS, version 9.4 (SAS institute, Cary, NC) and analyzed using Stata 18.0 (STATA, College Station, TX). Analysis coding is available at Github (https://github.com/yihuiyang2/ParentalSuicide). |

For manuscripts utilizing custom algorithms or software that are central to the research but not yet described in published literature, software must be made available to editors and reviewers. We strongly encourage code deposition in a community repository (e.g. GitHub). See the Nature Portfolio guidelines for submitting code & software for further information.

## Data

Policy information about availability of data

All manuscripts must include a data availability statement. This statement should provide the following information, where applicable:
- Accession codes, unique identifiers, or web links for publicly available datasets
- A description of any restrictions on data availability
- For clinical datasets or third party data, please ensure that the statement adheres to our policy

The Public Access to Information and Secrecy Act in Sweden prohibits individual-level data being publicly available. Researchers who are interested in replicating this study can apply for individual-level data from Total Population Register, Multi-Generation Register, and Longitudinal Integration Database for Health Insurance and Labor Market Studies through Statistics Sweden (https://www.scb.se/en/services/ordering-data-and-statistics/ordering-microdata/). Data on patient health

## Research involving human participants, their data, or biological material

Policy information about studies with human participants or human data. See also policy information about sex, gender (identity/presentation), and sexual orientation and race, ethnicity and racism.

| | |
|---|---|
| Reporting on sex and gender | This study analyzed sex-specific incidence rate of suicide attempt. Sex was determined based on Swedish Medical Birth Register and Total Population Register. |
| Reporting on race, ethnicity, or other socially relevant groupings | We collected information on participants' country of birth from Swedish Total Population Register, as a proxy of potential differential health care-seeking behaviour. |
| Population characteristics | In the study sample, mothers were younger, had a higher educational attainment yet a lower income, than the fathers (Table 1). In addition, mothers were more likely to have a history of depression and other psychiatric disorders or a history of suicide attempt. |
| Recruitment | We identified mothers from Swedish Medical Birth Register (MBR), which includes antenatal and obstetric records on 98% of all births in Sweden. Fathers were mainly identified from Multi-Generation Register, which provides information on familial linkages for individuals born since 1932 in Sweden. Because fathers were not reported for 2% of children in the MGR, we complemented the identification of fathers as partners of women who reported as cohabiting with "the-father-to-be" at their first antenatal visit, based on the MBR. Selection bias could emerge when participation is related to both exposure and outcome. In our study, exposure is sex and period, which should not be affected by participation. Therefore, the selection bias should be minimized in our study. |
| Ethics oversight | Swedish Ethical Review Authority |

Note that full information on the approval of the study protocol must also be provided in the manuscript.

# Field-specific reporting

Please select the one below that is the best fit for your research. If you are not sure, read the appropriate sections before making your selection.

☒ Life sciences ☐ Behavioural & social sciences ☐ Ecological, evolutionary & environmental sciences

For a reference copy of the document with all sections, see nature.com/documents/nr-reporting-summary-flat.pdf

# Life sciences study design

All studies must disclose on these points even when the disclosure is negative.

| | |
|---|---|
| Sample size | No sample size calculation was performed. We determined sample size based on predefined inclusion and exclusion criteria. Specifically, based on the MBR, we first identified 1,258,824 women and their 2,263,596 pregnancies during 2001-2021 in Sweden. After excluding women who had an invalid personal identification number, missing information on length of gestation, conflicting information (e.g., died before start of pregnancy or delivery date), and erroneous records, 1,236,816 women and 2,196,276 pregnancies remained in the analysis (Figure S1). We then identified fathers based on the Multi-Generation Register (MGR) . Because fathers were not reported for 2% of children in the MGR, we complemented the identification of fathers as partners of women who reported as cohabiting with "the-father-to-be" at their first antenatal visit, based on the MBR. Among the 1,213,034 fathers identified, we excluded those who had invalid personal identification numbers or conflicting information, leaving 1,175,674 fathers in the analysis (Figure S1). |
| Data exclusions | Among 1,258,824 mothers who gave birth during 2001-2021 in Sweden, we excluded women who had an invalid personal identification number, missing information on length of gestation or conflicting information (e.g., died before start of pregnancy or delivery date), leaving 1,236,816 mothers in the analysis. Among the 1,213,034 fathers identified, we excluded those who had invalid personal identification numbers or conflicting information, leaving 1,175,674 fathers in the analysis. |
| Replication | We have rerun the analysis coding several times, and the results consistently matched to what we presented in the manuscript. |
| Randomization | This is an observational study and no randomization was performed. |
| Blinding | This is an observational study and no randomization was performed. |

# Reporting for specific materials, systems and methods

We require information from authors about some types of materials, experimental systems and methods used in many studies. Here, indicate whether each material, system or method listed is relevant to your study. If you are not sure if a list item applies to your research, read the appropriate section before selecting a response.

## Materials & experimental systems

| n/a | Involved in the study |
|-----|----------------------|
| ☒ | Antibodies |
| ☒ | Eukaryotic cell lines |
| ☒ | Palaeontology and archaeology |
| ☒ | Animals and other organisms |
| ☒ | Clinical data |
| ☒ | Dual use research of concern |
| ☒ | Plants |

## Methods

| n/a | Involved in the study |
|-----|----------------------|
| ☒ | ChIP-seq |
| ☒ | Flow cytometry |
| ☒ | MRI-based neuroimaging |

## Plants

| | |
|---|---|
| Seed stocks | *Report on the source of all seed stocks or other plant material used. If applicable, state the seed stock centre and catalogue number. If plant specimens were collected from the field, describe the collection location, date and sampling procedures.* |
| Novel plant genotypes | *Describe the methods by which all novel plant genotypes were produced. This includes those generated by transgenic approaches, gene editing, chemical/radiation-based mutagenesis and hybridization. For transgenic lines, describe the transformation method, the number of independent lines analyzed and the generation upon which experiments were performed. For gene-edited lines, describe the editor used, the endogenous sequence targeted for editing, the targeting guide RNA sequence (if applicable) and how the editor was applied.* |
| Authentication | *Describe any authentication procedures for each seed stock used or novel genotype generated. Describe any experiments used to assess the effect of a mutation and, where applicable, how potential secondary effects (e.g. second site T-DNA insertions, mosiacism, off-target gene editing) were examined.* |

