## [Peer Review File · Nature Human Behaviour]

Sex difference in parental risk of suicide attempt during and after pregnancy in Sweden

Corresponding Author: Ms Yihui Yang

Version 0:

Decision Letter:

19th November 2024

Dear Ms Yang,

Thank you once again for your manuscript, entitled "Sex difference in parental risk of suicide attempt during and after pregnancy: a nationwide register-based study," and for your patience during the peer review process.

Your manuscript has now been evaluated by 3 reviewers, whose comments are included at the end of this letter. Although the reviewers find your work to be of interest, they also raise some important concerns. We are interested in the possibility of publishing your study in *Nature Human Behaviour*, but would like to consider your response to these concerns in the form of a revised manuscript before we make a decision on publication.

To guide the scope of the revisions, the editors discuss the referee reports in detail within the team, including with the chief editor, with a view to (1) identifying key priorities that should be addressed in revision and (2) overruling referee requests that are deemed beyond the scope of the current study. We hope that you will find the prioritised set of referee points to be useful when revising your study. Please do not hesitate to get in touch if you would like to discuss these issues further.

Specifically, please address the following:

1. Reviewer #3 suggested additional analysis to investigate the effect excluding week 53. Please include this as sensitivity analysis, and modify your results as needed.
2. Our reviewers also raised concerns (including in their comments to editors) about the appropriateness of the analytical methods and interpretations. Please provide our reviewers with access to a dataset containing sufficiently granular data for them to evaluate the robustness of the work, as well as access to the code used for your analyses.
3. Reviewer #3 raised concerns about mechanism underlying the observed decline in incident rates for both mothers and fathers during the 12-month leadup to conception, as well as that for the modest correlation between mothers and fathers' incident rates, please address this issue with thorough discussion.

In sum, we invite you to revise your manuscript taking into account all reviewer and editor comments. We are committed to providing a fair and constructive peer-review process. Do not hesitate to contact us if there are specific requests from the reviewers that you believe are technically impossible or unlikely to yield a meaningful outcome.

We hope to receive your revised manuscript within two months. I would be grateful if you could contact us as soon as possible if you foresee difficulties with meeting this target resubmission date.

- Include a "Response to the editors and reviewers" document detailing, point-by-point, how you addressed each editor and referee comment. If no action was taken to address a point, you must provide a compelling argument. When formatting this document, please respond to each reviewer comment individually, including the full text of the reviewer comment verbatim followed by your response to the individual point. This response will be used by the editors to evaluate your revision and sent back to the reviewers along with the revised manuscript.

- Highlight all changes made to your manuscript or provide us with a version that tracks changes.

Link Redacted

We look forward to seeing the revised manuscript and thank you for the opportunity to review your work. Please do not hesitate to contact me if you have any questions or would like to discuss these revisions further.

Sincerely,

[Redacted Signature]

Nature Human Behaviour

Reviewer expertise:

Reviewer #1: Parenthood suicide attempt, Perinatal Mental Health

Reviewer #2: Perinatal Mental Health

Reviewer #3: Suicide, Perinatal Mental Health

REVIEWER COMMENTS:

Reviewer #1 (Remarks to the Author):

The study is original and of great relevance to the mental health of parents. The authors conducted a study based on birth records, which allowed the monitoring of individuals and the incidence of suicide attempts. The authors suggest that there may be a protective effect of childbirth against suicide attempts among mothers and fathers, which corroborates the existing literature. I consider some small points to be clarified:

- Method: Were all people alive in Sweden in 2021 included? This is unclear - Results: Was there a mother or a father at age 10? What is the reference for quintile 1 in income? Is it the highest or lowest income? Diagnosis of a previous psychiatric disorder made in primary care: describe by whom it was made and whether it was a clinical interview or some instrument. - Discussion: In line 269: "We speculate that healthier and more economically advantaged individuals may have been more likely to choose to become pregnant during this period than those who were less advantaged." Is this a hypothesis for the result, or is there actually evidence?

Reviewer #2 (Remarks to the Author):

The manuscript deals with the sex difference of suicide attempt around childbirth. As it is rightly noted, in general population the risk of suicide attempt is bigger for women. The authors find in comparison to the control group of preconception period that during the childbirth and within the year after it the risk of paternal suicide attempt is bigger than the maternal one. It is a novel outcome and for that reason very valuable research and has a new outlook for policy recommendations.

The article is based on the register-based data of Sweden, which enables to detect through linkages to multiple other registries the full population under concern over the years 2001-2021, enabling to understand also the trends in this phenomena. The data is well organised, the covariates are chosen appropriately and age standardization makes the results valid for the conclusions made. In the analytical approach we value highly the usage of standardised incidence rates. We also appreciate the secondary analyses of the phenomena by high risk age groups. As the results of higher paternal risk of suicide attempt during childbirth and especially in the second part of the postpartum year hold true for the whole time period and for separately analysed age group, the article presents a very valuable piece of evidence to take these results into account in perinatal care, dealing with fathers. Therefore we find the article novel and valuable both from the angle of highlighting the issue also for European context, analysing the related time trends (some hope for decreasing trends), high risk age groups and the periods related to childbirth and postpartum in particular.

Reviewer #3 (Remarks to the Author):

<Key results: Please summarise what you consider to be the outstanding features of the work.>

This work reports the incident rate of suicide attempt is lower for mothers than fathers during the periods i) pregnancy and ii) the first 12 months of the antepartum period. This is a reversal of the relative trends seen in the 12 month period leading up to conception, where fathers display a higher incident rate than mothers.

<Validity: Does the manuscript have flaws which should prohibit its publication? If so, please provide details.>
I have some suggestions for modest improvements to be made before publication

<Originality and significance: >

My expertise is primarily in statistical methods rather than the subject matter, so I refrain from making comment on this question.

<Data & methodology: Please comment on the validity of the approach, quality of the data and quality of presentation. Please note that we expect our reviewers to review all data, including any extended data and supplementary information. Is the reporting of data and methodology sufficiently detailed and transparent to enable reproducing the results?>

I consider the approach sound and believe the data to be of high quality. Some minor comments regarding the presentation are as follows:

- a) please expand on the method of Imputation of delivery date in the eMethods to clarify the process used. The sentence beginning 'If the length... on line 35, ending ...independent database on line 37 does not explain why the median length of stay was added to the date of admission to determine delivery date. I understand this to imply that imputed delivery dates are assumed to occur at the end of the stay.
- b) The totals of PYs in Table 1 are consistent with the breakdowns by characteristic categories in Table 1, but they are NOT consistent with the total of PYs listed in Table S2. The IRs quoted in the first paragraph of the results use the quantities listed in Table S2. Please reconcile.
- c) Please state the smoothing bandwidth parameters used in in Figure 1, Figure 2 and Supplementary Figures S2-S6.
- d) Please elaborate on the method of standardisation by age group and calendar period referenced in the figures. In the methods section only distribution by age group was described.
- e) Please indicate what the shaded areas are in each graph. – E.g. xx% confidence bounds and, in the methods how they were calculated
- f) The horizontal reference line in Figure 3 would be better coloured black. The current colour of blue leads the reader to associate it exclusively with fathers.
- g) If possible, please standardise the y axis scales across preconception, antepartum and postpartum graphs in Supplementary figures S2, S7, S9
- h) In reference to knots used in restricted cubic splines, I assume "5%, 35%, 65% and 95% of distribution of events" corresponds to 5%, 35%, 65% and 95% of the number of weeks on the relevant x axes. Please relabel as similar, to aid in reader comprehension in Figure 3, Figure 4 and Supplementary Figures S7-S9.
- i) Please state adjustment variables used in estimating IRR (currently stated for IRDs only) in Statistical analysis in methods.
- j) Please expand on the parameterisation of the assumed Poisson regression models for IRR and IRD and report parameter results. (In supplementary materials would be ok). It is apparent that the results for IRRs given in Supplementary tables S3 and S4 tabulate the cubic spline smoothed estimates.

<preregistration: >

No pre-registration

<appropriate use of statistics and treatment of uncertainties: >

Statistical methods are appropriate conditional on concern a) being addressed.

a) I do not understand why the preconception incident rate ("baseline") is calculated for each of 52 weeks. This implies there is specific interest in the way incident rate during this period changes over time (e.g. see b) below), and that the relationship between the incident rate of a given week of either pregnancy or postpartum is related to the incident rate of the exact corresponding week in preconception.

If so, this needs to be explained in the methods.

If not, I believe it would be better (and remove noise associated with random fluctuations in the preconception period) to compare each week of pregnancy and the postpartum period to a single baseline estimate for each of mothers and fathers from the preconception period. There could be several valid estimates of such baseline estimates; any reasonable well defined method would suffice.

b) It is interesting to note the apparent decline in incident rates (and standardised incident rates) for both mothers and fathers during the 12 month leadup to conception. Is there an explanation for this? Is there a similar pattern in the 12 months prior to the beginning of the 12 month leadup to conception (ie beginning 2 years before conception)? It is also surprising to note a modest correlation ($r=0.5$) between the incident rates for mothers and fathers across the 53 weeks of preconception. Was this expected by the authors?

<Custom code: If the work includes custom code, does the code run as intended? If you are unable to access the code, please contact us. >

No custom code. I believe all analyses would have been run with standard code accessible in the statistical packages referenced.

<Conclusions:> Do you find that the conclusions and data interpretation are robust, valid and reliable?

Conclusions and data interpretation appear valid. However, when considering the raw data in Figure 2, the following interpretation may be considered: Fathers' suicide attempt incident rate is relatively constant from the beginning of the preconception period to the end of the postpartum period, with the exception of a dip during the period approximately 5-10 weeks gestation to approximately 5-10 weeks postpartum. Coinciding with this, mothers suicide attempt incident rate drops markedly from the beginning of pregnancy and begins to rise at approximately 6 weeks postpartum before stabilising at approximately 6 months postpartum to a level which remains much lower than mothers' preconception levels and fathers' levels at the end of postpartum. Given the apparent unexplained decreasing trend for IRs in Figure 2(Preconception), the variation introduced by comparing antepartum and postpartum rates to fluctuating preconception rates, caution surrounding estimates from weeks 53 and the arbitrary nature of cubic splines, the IRR trends displayed in Figure 3 may have accumulated distortions. I would caution against conclusions that refer to specific time points e.g. "an increased risk from the 37th week postpartum". Firstly, the 'risk' is the average risk, estimated from the combination of several modelling choices. Secondly, there is no obvious step function at 37 weeks, so I would suggest precision no more than 'at around 37 weeks' .

I am puzzled by the results presented in Figure 4, Preconception. I agree with the conclusion from the data presented in Figure 4—that is “Compared to fathers, mothers had a similar risk of suicide attempt throughout the preconception period”. However, this is at odds with the precis of the introduction “in the general population, women are more likely than men to attempt suicide”, the general pattern observed in Figure 1 (Preconception), and the statement in the Results paragraph headed Secular trends in parental suicide attempt, namely “Compared to fathers, mothers had higher incidence rates of suicide attempt in the preconception period...”.

<Suggested improvements: Please list additional analyses, experiments or data that could help strengthening the work in a revision.>

I suggest investigating the effect/s, if any of excluding week 53 from analyses. This is due to the much smaller number of PYs available, and the apparent ‘kick-up’ in the SIR for fathers for week 53 postpartum (Figure 2).

<References: Does this manuscript reference previous literature appropriately? If not, what references should be included or excluded?>

Statistical literature is appropriate

<Clarity and context: Is the abstract clear, accessible? Are abstract, introduction and conclusions appropriate?>

Yes

<please indicate any particular part of the manuscript, data, or analyses that you feel is outside the scope of your expertise, or that you were unable to assess fully.>

My expertise is primarily in statistical methods rather than the subject matter, so I restricted my assessment to those areas.

<it would be most helpful if you could take a look at the code provided by the authors and comment on whether it provides sufficiently clear information and documentation that would make it possible to test the code and replicate the paper's findings. We would also be grateful if you, or someone in your group, could look at the code more closely and verify that it can indeed be run.>

No code was supplied

Version 1:

Decision Letter:

26th February 2025

Dear Ms Yang,

Thank you once again for your revised manuscript, entitled "Sex difference in parental risk of suicide attempt during and after pregnancy: a nationwide register-based study," and for your patience during the re-review process.

Your manuscript has now been evaluated by Reviewer #3 from the original round of review. Please note that, as Reviewer #1 was unable to re-review at this time, Reviewer 3 commented on responses to Reviewer 1's concerns. All reviewer feedback is included at the end of this letter. Although the reviewers found your manuscript to have improved during revision, they also raise some important outstanding concerns. We remain very interested in the possibility of publishing your study in *Nature Human Behaviour*, but would like to consider your response to these outstanding concerns in the form of a revised manuscript before we make a decision on publication.

Specifically, please address the following:

- Please consider Reviewer #3's suggestion of using single estimates from the pre-conception period as default analysis and the current analyses with weekly pre-conception data as supplementary, provide full and appropriate justification for your chosen analytical approach, and present the results of the alternative analysis in your supplementary material.

- Reviewer #3 also raised concerns about the cubic splines analysis. Please add the full code you used for cubic splines to the github repository, explain in more depth about this analysis, in particular, respond to Reviewer #3's concern about the choice you made to use the same three knots for all three periods. Consider adding LOWESS smoothing there to be consistent.

In sum, we invite you to revise your manuscript taking into account all reviewer and editor comments. We are committed to providing a fair and constructive peer-review process. Do not hesitate to contact us if there are specific requests from the reviewers that you believe are technically impossible or unlikely to yield a meaningful outcome.

We hope to receive your revised manuscript within 4-8 weeks. I would be grateful if you could contact us as soon as possible if you foresee difficulties with meeting this target resubmission date.

- Include a "Response to the editors and reviewers" document detailing, point-by-point, how you addressed each editor and referee comment. If no action was taken to address a point, you must provide a compelling argument. This response will be used by the editors and reviewers to evaluate your revision.
- Highlight all changes made to your manuscript or provide us with a version that tracks changes.

Link Redacted

We look forward to seeing the revised manuscript and thank you for the opportunity to review your work. Please do not hesitate to contact me if you have any questions or would like to discuss these revisions further.

Sincerely,

[Redacted Signature]

Nature Human Behaviour

Reviewer expertise:

Reviewer #3: Suicide, Perinatal Mental Health

REVIEWER COMMENTS:

Reviewer #3 (Remarks to the Author):

Point a)

I think my misunderstanding was due to the omission of the word "half". If I understand correctly, consider where imputation was required for a birth to a mother of a given parity and delivery method. Let us assume that the median length of stay for all births of the same parity and delivery method in the independent database was N days, then the best estimate of delivery date would be admission date plus $N/2$ (half way between admission and discharge). (If you added N to the admission date, you would get to the typical discharge date). It is evident that you have made the calculation as $median/2$ – it was just the omission of "half" in the text which caused confusion.

There is no need for the addition of the text "., since the delivery date must come on or after the admission date, and we assumed that the delivery happened in the middle of the stay."

Point b)

thank you for the clarification.

Point c)

Thank you for noting this. I now understand that you applied LOWESS to the 95% SIR confidence bounds. My query in e) was due to the misunderstanding that you had smoothed SIR by LOWESS, and then calculated the 95% bounds of the LOWESS.

Point d) Point e)

Thank you for these clarifications.

Point f)

Thank you for this change

Point g)

Thank you for standardizing S7 and S9. S2 appears to have different scales, with 1 at different heights above the x axis.

Point h)

Thank you. Please also correct this in the methods.

Point i)

Thank you

Point j)

I remain unclear of the process used to incorporate cubic splines. The code for their estimation is not included in the github repository. Thank you for including the parameters listed. However, they suggest the same knots were used for all three periods. Given the pregnancy period is only 40 weeks, compared to 52 weeks for the pre-conception and postpartum periods, this infers inconsistency. In addition, the 95% confidence bounds on related Figures (e.g. Figure 3 and figure 4, others in supplementary) appear quite narrow, considering the variability of the raw data (Figure 1 and Figure 2). Perhaps the standard errors of the smoothed spline estimates have not been incorporated into the total variation after adjustment for country of birth, age, etc?

I understand the desire to smooth results to aid the reader in following broad patterns more easily, but I am not convinced that the given application of cubic splines is the best way to do so in this instance. As you show in Figure S14 and figure S15, the results without splines are consistent. There is more noise in the figure, but this is more realistic and less likely for the reader to infer more regularity in the pattern than we see. LOWESS smoothing could be repeated here

Statistical methods, Point a)

I appreciate the authors comparing IRR rates in antepartum and postpartum to an average of the preconception period. In my view,

this simplified approach is much more accessible to the reader. I would prefer this as the default analysis. I do not follow how comparing weekly rates would help to alleviate seasonal variation as the calculated rates for each week are an aggregation of hundreds of thousands of persons, presumably measured over the full range of seasons. If the authors are happy that the small significant declines in SIR for the preconception period (Table R1) are of negligible clinical concern, I strongly advocate that single measures of preconception suicide rate (one each for mothers and fathers) are used as reference measures when comparing weekly pregnancy rates and weekly postpartum rates.

In other words, either there is a decreasing trend in the preconception period, which justifies comparison of rates in each week of pregnancy to rates exactly 12 months before, OR there is no evidence of clinically relevant decline in preconception period, in which case it makes most sense to compare weekly pregnancy rates with a single preconception estimate.

Other responses

thank you for all other clarifications.

I confirm that I consider the overall end conclusions about the data to be sound. My personal preference is simplified methods (single pre-conception estimates, and minimal use of smoothing).

Reviewer #3 (Remarks on code availability):

the code is clear. however there is no code covering the calculation of cubic splines. the given code accesses the generated spline estimates from a data file, and does not appear to incorporate the standard errors of these estimates into the overall uncertainty of the model.

Version 2:

Decision Letter:

Our ref: NATHUMBEHAV-24072768B

14th May 2025

Dear Dr. Yang,

Thank you for submitting your revised manuscript "Sex difference in parental risk of suicide attempt during and after pregnancy: a nationwide register-based study" (NATHUMBEHAV-24072768B). It has now been editorially evaluated, and we find that the paper has improved in revision. We will therefore be happy in principle to publish it in Nature Human Behaviour, pending minor revisions to comply with our editorial and formatting guidelines.

We are now performing detailed checks on your paper and will send you a checklist detailing our editorial and formatting requirements within two weeks. Please do not upload the final materials and make any revisions until you receive this additional information from us.

Sincerely,

Nature Human Behaviour

Version 3:

Decision Letter:

Dear Ms Yang,

We are pleased to inform you that your Article "Sex difference in parental risk of suicide attempt during and after pregnancy in Sweden", has now been accepted for publication in Nature Human Behaviour.

Authors may need to take specific actions to achieve compliance with funder and institutional open access mandates. If

your research is supported by a funder that requires immediate open access (e.g. according to [Plan S principles](https://www.springernature.com/gp/open-science/plan-s-compliance) or the [NIH public access policy](https://www.springernature.com/gp/open-science/us-federal-agency-compliance)) then you

should select the gold OA route, and we will direct you to the compliant route where possible. Because authors warrant under our subscription licensing terms that they haven't committed to licensing any version of their article under a licence inconsistent with the terms of our agreement – including the applicable embargo period – publication under the subscription model isn't suitable for authors whose funders require no embargo.

With best regards,

Editor
Nature Human Behaviour

P.S. Click on the following link if you would like to recommend Nature Human Behaviour to your librarian
<http://www.nature.com/subscriptions/recommend.html#forms>

** Visit the Springer Nature Editorial and Publishing website at http://editorial-jobs.springernature.com?utm_source=eJP_NHumB_email&utm_medium=eJP_NHumB_email&utm_campaign=eJP_NHumB for more information about our career opportunities. If you have any questions please click [here](mailto:editorial.publishing.jobs@springernature.com).

Open Access This Peer Review File is licensed under a Creative Commons Attribution 4.0 International License, which permits use, sharing, adaptation, distribution and reproduction in any medium or format, as long as you give appropriate credit to the original author(s) and the source, provide a link to the Creative Commons license, and indicate if changes were made. In cases where reviewers are anonymous, credit should be given to 'Anonymous Referee' and the source. The images or other third party material in this Peer Review File are included in the article's Creative Commons license, unless indicated otherwise in a credit line to the material. If material is not included in the article's Creative Commons license and your intended use is not permitted by statutory regulation or exceeds the permitted use, you will need to obtain permission directly from the copyright holder.

19th November 2024

Dear Ms Yang,

Thank you once again for your manuscript, entitled "Sex difference in parental risk of suicide attempt during and after pregnancy: a nationwide register-based study," and for your patience during the peer review process.

Your manuscript has now been evaluated by 3 reviewers, whose comments are included at the end of this letter. Although the reviewers find your work to be of interest, they also raise some important concerns. We are interested in the possibility of publishing your study in *Nature Human Behaviour*, but would like to consider your response to these concerns in the form of a revised manuscript before we make a decision on publication.

To guide the scope of the revisions, the editors discuss the referee reports in detail within the team, including with the chief editor, with a view to (1) identifying key priorities that should be addressed in revision and (2) overruling referee requests that are deemed beyond the scope of the current study. We hope that you will find the prioritised set of referee points to be useful when revising your study. Please do not hesitate to get in touch if you would like to discuss these issues further.

Response: We thank the editor for the positive feedback on our work.

Specifically, please address the following:

1. Reviewer #3 suggested additional analysis to investigate the effect excluding week 53. Please include this as sensitivity analysis, and modify your results as needed.

Response: Thank you for this helpful suggestion. We have now excluded week 53 from the analysis. Although the IRRs comparing the antepartum and postpartum to the preconception period, and those comparing mothers to fathers in the preconception and postpartum period, changed minimally (**Figure S12** and **S13**), the rise in paternal absolute IR in the late postpartum period was no longer noted (**Figure S11** below).

Figure S11 Standardized incidence rate of parental suicide attempt before, during, and after pregnancy by week, after excluding week 53 from the analysis

The follow-up during pregnancy was censored at week 40 due to few subsequent events. Incidence rates were standardized by distribution of age group and calendar period of the accumulated person-years during follow-up. Locally Weighted Scatterplot Smoothing (bandwidth=0.8) was used to estimate the trend (solid line), while the dots indicate the weekly incidence rates. The shaded areas indicate 95% confidence interval.

Figure S12 Incidence rate ratio of parental suicide attempt during and after pregnancy when compared with the corresponding week before pregnancy, after excluding week 53 from the analysis

The follow-up during pregnancy was censored at week 40 due to few subsequent events. A restricted cubic spline with 4 knots, placed at 5th percentile, 35th percentile, 65th percentile and 95th percentile of distribution of weeks on the relevant x axis, was used to estimate the incidence rate ratios. The incidence rate ratios were adjusted for country of birth, age, calendar year, education level, civil status, category of income, primiparity, history of psychiatric disorders and history of suicide attempt, all derived at the start of each period. The shaded areas indicate 95% confidence interval.

Figure S13 Incidence rate ratio of suicide attempt among mothers when compared with the corresponding week among fathers in preconception and postpartum period, after excluding week 53

A restricted cubic spline with 4 knots, placed at 5th percentile, 35th percentile, 65th percentile and 95th percentile of distribution of weeks on the relevant x axis, was used to estimate the incidence rate ratios. The incidence rate

ratios were adjusted for country of birth, age, calendar year, education level, civil status, category of income, primiparity, history of psychiatric disorders and history of suicide attempt, all derived at the start of each period. The shaded areas indicate 95% confidence interval.

We made changes in the manuscript accordingly, as described below.

Modified text:

Page 17: In addition, due to the small number of person-years, we excluded week 53 from analyses, and calculated SIR by week, and estimated IRRs comparing antepartum and postpartum to preconception, and comparing mothers to fathers in preconception and postpartum.

Page 7: Furthermore, when excluding week 53 from the analysis, the rise in paternal absolute IR in late postpartum period was less pronounced yet the CIs overlapped with those from the primary analysis (Figure S11) whereas the IRRs changed minimally (Figure S12-S13).

2. Our reviewers also raised concerns (including in their comments to editors) about the appropriateness of the analytical methods and interpretations. Please provide our reviewers with access to a dataset containing sufficiently granular data for them to evaluate the robustness of the work, as well as access to the code used for your analyses.

Response: Thank you. We understand the concern of the editors and reviewers. However, the Public Access to Information and Secrecy Act in Sweden prohibits making individual-level data publicly available due to concerns of back tracing. Regardless, we have explored the possibility of sharing group-level data (e.g., number of events and person time by levels of covariates); however, the multiplicative combination of multiple covariates, especially week which has 53 levels, result in millions of groups. Together with the rareness of parental suicide attempt, we have very few events, i.e., less than 5, in majority of the groups. Sharing such aggregated data would be against the Swedish legislation considering they are potentially traceable to a specific individual. However, we have uploaded statistical codes and log files to Github (<https://github.com/yihuiyang2/ParentalSuicide>) for editors and reviewers to evaluate the robustness of our analytic approach.

3. Reviewer #3 raised concerns about mechanism underlying the observed decline in incident rates for both mothers and fathers during the 12-month leadup to conception, as well as that for the modest correlation between mothers and fathers' incident rates, please address this issue with thorough discussion.

Response: These points are well-taken.

1) We would like to clarify that the curves were smoothed by LOWESS, and there is no obvious trend of the scattered dots from visual inspection. We also formally tested the magnitude of the trend by week using Poisson regression. When analyzing the association between week and risk of suicide attempt in the preconception year, we found that the incidence rate remained largely unchanged over weeks (**Table R1**). Although the effect sizes are statistically significant, it is possibly due to our large sample size, which makes it oversensitive to minor deviation from the null hypothesis. We also performed the piecewise Poisson regressions to detect potential non-linear associations, but these also yielded very stable incidence rates across different periods in the preconception year (**Table R1**). In addition, we also did linear regression, by using the age and calendar year-standardized

incidence rates per 1000 person years as dependent variable, and week (continuous variable) as independent variable. The results also showed no statistically significant trend.

Table R1 Association between week and incidence rate of suicide attempt in the preconception year

	Mothers		Fathers	
	IRR (95% CI)	P-value	IRR (95% CI)	P-value
Poisson regression	1.00 (0.99-1.00)*	<0.001	1.00 (0.99-1.00)*	<0.001
Piecewise Poisson regression ^a				
week ≤15	0.98 (0.97-0.99)*	0.005	0.99 (0.98-1.00)	0.182
week >15 and ≤35	1.00 (1.00-1.01)	0.249	1.00 (0.99-1.01)	0.793
week >35	0.99 (0.98-1.00)*	0.031	0.99 (0.98-1.00)	0.117
Linear regression	-0.01 (-0.01-0.00)	0.051	-0.01 (-0.02--0.00)*	<0.001

CI, confidence interval; IRR, incidence rate ratio.

Poisson regression was adjusted for week (continuous variable), and groups of age and calendar period. Linear regression was performed by using age and calendar year-standardized incidence rates per 1000 person years as dependent variable, and week (continuous variable) as independent variable.

^aThe cutoff was chosen based on visual inspection of turning points indicated in Figure 2.

We have also discussed the trend in preconception period in the manuscript.

Modified text:

Page 9: Sex difference in preconceptional suicide attempt

Preconception is considered as a relative healthy period when expectant mothers plan for pregnancy⁴⁹, and possibly for fathers as well. Our study observed a largely stable, given the wide CIs, SIR of suicide attempt over the weeks in preconception period, compared to the trend during pregnancy. This suggests such “health bias” may not significantly affect the weekly trend of suicide attempts before pregnancy.

2) We are not aware of studies showing a correlation between expectant parents’ suicide behaviour specifically in the preconception period, but some literature has pointed out associations between suicide between spouses regardless of their life stage ^[1,2], possibly due to the stress brought by suicide of a partner, shared environmental factors between the cohabitating partners or selective mating. Therefore such association was expected and discussed in the manuscript.

Modified text:

Page 9: Sex difference in preconceptional suicide attempt

In addition, we found a similar risk of suicide attempt among mothers and fathers over weeks throughout the preconception period. The trend of weekly SIRs in preconception period among fathers somewhat echoed the trend among mothers. This is consistent with previous literature reporting a positive link on suicide risk between spouses, possibly due to stress brought by suicide of a partner, or shared environmental factors between the cohabitating partners or selective mating^{50,51}.

References:

1. Agerbo E. Risk of suicide and spouse's psychiatric illness or suicide: nested case-control study. *BMJ*. 2003;327(7422):1025-1026.
2. Jang J, Park SY, Kim YY, et al. Risks of suicide among family members of suicide victims: A nationwide sample of South Korea. *Front Psychiatry*. 2022;13:995834. doi:10.3389/fpsy.2022.995834

Response: Thank you. We have formatted the manuscript according to journal's requirement.

In sum, we invite you to revise your manuscript taking into account all reviewer and editor comments. We are committed to providing a fair and constructive peer-review process. Do not hesitate to contact us if there are specific requests from the reviewers that you believe are technically impossible or unlikely to yield a meaningful outcome.

We hope to receive your revised manuscript within two months. I would be grateful if you could contact us as soon as possible if you foresee difficulties with meeting this target resubmission date.

- **Include a "Response to the editors and reviewers" document detailing, point-by-point, how you addressed each editor and referee comment. If no action was taken to address a point, you must provide a compelling argument. When formatting this document, please respond to each reviewer comment individually, including the full text of the reviewer comment verbatim followed by your response to the individual point. This response will be used by the editors to evaluate your revision and sent back to the reviewers along with the revised manuscript.**
- **Highlight all changes made to your manuscript or provide us with a version that tracks changes.**

We look forward to seeing the revised manuscript and thank you for the opportunity to review your work. Please do not hesitate to contact me if you have any questions or would like to discuss these revisions further.

Sincerely,

██████████
██████████

Nature Human Behaviour

Reviewer expertise:

Reviewer #1: Parenthood suicide attempt, Perinatal Mental Health

Reviewer #2: Perinatal Mental Health

Reviewer #3: Suicide, Perinatal Mental Health

REVIEWER COMMENTS:

Reviewer #1 (Remarks to the Author):

The study is original and of great relevance to the mental health of parents. The authors conducted a study based on birth records, which allowed the monitoring of individuals and the incidence of suicide attempts. The authors suggest that there may be a protective effect of childbirth against suicide attempts among mothers and fathers, which corroborates the existing literature. I consider some small points to be clarified:

Response: Thank you for the positive comments.

- Method: Were all people alive in Sweden in 2021 included? This is unclear

Response: Our study population was defined as all women who had pregnancies during 2001-2021, as well as their spouses in Sweden. After identifying the eligible individuals using the Swedish Medical Birth Register and the other population-based registers, we excluded those who had invalid personal identification number, missing information on length of gestation, conflicting information, or erroneous records (page 13-14 in the manuscript).

Therefore, women who were alive in 2021 but had no pregnancies during 2001-2021 and their spouses were not included. In addition, women who were alive in 2021 and had pregnancies during 2001-2021, but satisfied the exclusion criteria, were also excluded together with their spouses. Since our inclusion and exclusion criteria did not rely on individuals' living status in 2021, we did not state it in the manuscript. Study participants who died during the study period were censored on that date.

- Results: Was there a mother or a father at age 10?

Response: Thank you for raising this question. First, we would like to correct the minimum age in our data due to the wrong labeling in Table 1. Over a study period of 21 years in the entire country, there are few mothers (< 5) who gave birth at age 13, translating to an age of 11 at the start of their preconception year. There was no father at age 10. We have now rectified this in the manuscript accordingly.

What is the reference for quintile 1 in income? Is it the highest or lowest income?

Response: We now clarify in the Methods section and in the footnote of Table 1 that quintile 1 is the lowest, i.e. corresponds to the 0-20th percentile of the income distribution.

Modified text:

Page 14: Proxies of socioeconomic status, including education level and income, were obtained from the Longitudinal Integration Database for Health Insurance and Labor Market Studies (LISA). Income was further classified into 5 quantiles, with quintile 1 representing the lowest, i.e. the 0-20 percentile of the income distribution.

Table 1, table note: Quantile 1, 2, 3, 4 and 5 represent 0-20th percentile (lowest), 21-40th percentile, 41-60th percentile, 61-80th percentile and 81-100th percentile (highest) of the income distribution, respectively.

Diagnosis of a previous psychiatric disorder made in primary care: describe by whom it was made and whether it was a clinical interview or some instrument.

Response: Thank you for the question. First, we would like to clarify that we identified diagnoses of psychiatric disorder from the National Patient Register, which contains clinical diagnoses made by specialists and has been shown to have generally a good validity^[1-4]. Since many psychiatric disorders are commonly diagnosed in primary care, we also used information on prescribed psychiatric medications from the Prescribed Drug Register to supplement diagnoses given in primary care (please see page 15 in the manuscript).

Regarding diagnostics, in Sweden, the national guidelines recommended that diagnosing psychiatric disorders should include (semi)-structured diagnostic interviews, e.g., the Mini International Neuropsychiatric Interview (MINI) and Structured Clinical Interview for DSM-IV-Axis I Disorders (SCID-I)^[5]. Previous studies suggest that the validity of the diagnoses of psychiatric disorders in primary care is good^[6]. Such information has been added to the manuscript.

Modified text:

Page 15: The national guideline in Sweden recommends that (semi)-structured diagnostic interviews should be carried out when diagnosing psychiatric disorders⁵⁴. We identified history of any psychiatric disorder using ICD-8/9/10 codes from NPR; to supplement with psychiatric diagnoses made in primary care by general practitioners, we also retrieved information on filled psychotropic prescriptions using Anatomical Therapeutic Chemical (ATC) codes (Table S1). Previous studies suggest that the validity of the diagnoses of psychiatric disorders in NPR and primary care is good⁵⁵⁻⁵⁹.

Reference:

1. Fazel S, Wolf A, Chang Z, Larsson H, Goodwin GM, Lichtenstein P. Depression and violence: a Swedish population study. *Lancet Psychiatry*. 2015;2(3):224-232. doi:10.1016/S2215-0366(14)00128-X
2. Meier SM, Petersen L, Mattheisen M, Mors O, Mortensen PB, Laursen TM. Secondary depression in severe anxiety disorders: a population-based cohort study in Denmark. *Lancet Psychiatry*. 2015;2(6):515-523. doi:10.1016/S2215-0366(15)00092-9

3. Kendler KS, Maes HH, Sundquist K, Ohlsson H, Sundquist J. Genetic and family and community environmental effects on drug abuse in adolescence: a Swedish national twin and sibling study. *Am J Psychiatry*. 2014;171(2):209-217. doi:10.1176/appi.ajp.2013.12101300
4. Hollander AC, Askegård K, Iddon-Escalante C, Holmes EA, Wicks S, Dalman C. Validation study of randomly selected cases of PTSD diagnoses identified in a Swedish regional database compared with medical records: is the validity sufficient for epidemiological research? *BMJ Open*. 2019;9(12):e031964. doi:10.1136/bmjopen-2019-031964
5. Vård vid depression och ångestsyndrom: huvudrapport med förbättringsområden: nationella riktlinjer - utvärdering 2019. Socialstyrelsen; 2019.
6. Sundquist J, Ohlsson H, Sundquist K, Kendler KS. Common adult psychiatric disorders in Swedish primary care where most mental health patients are treated. *BMC Psychiatry*. 2017;17:235. doi:10.1186/s12888-017-1381-4

- Discussion: In line 269: “We speculate that healthier and more economically advantaged individuals may have been more likely to choose to become pregnant during this period than those who were less advantaged.” Is this a hypothesis for the result, or is there actually evidence?

Response: Thank you. This is a hypothesis based on previous studies. We have clarified our reasoning in the manuscript.

Modified text:

Page 8: In addition, childbirth rate in Sweden has declined since 2010²⁶, partly due to financial uncertainty and labor market insecurity²⁷. We hypothesize that healthier and more economically advantaged individuals may have been more likely to choose to become pregnant during this period than those who are less advantaged.

Reviewer #2 (Remarks to the Author):

The manuscript deals with the sex difference of suicide attempt around childbirth. As it is rightly noted, in general population the risk of suicide attempt is bigger for women. The authors find in comparison to the control group of preconception period that during the childbirth and within the year after it the risk of paternal suicide attempt is bigger than the maternal one. It is a novel outcome and for that reason very valuable research and has a new outlook for policy recommendations.

The article is based on the register-based data of Sweden, which enables to detect through linkages to multiple other registries the full population under concern over the years 2001-2021, enabling to understand also the trends in this phenomena. The data is well organised, the covariates are chosen appropriately and age standardization makes the results valid for the conclusions made. In the analytical approach we value highly the usage of standardised incidence rates. We also appreciate the secondary analyses of the phenomena by high risk age groups. As the results of higher paternal risk of suicide attempt during childbirth and especially in the second part of the postpartum year hold true for the whole time period and for separately analysed age group, the article presents a very valuable piece of evidence to take these results into account in perinatal care, dealing with fathers. Therefore we find the article novel and valuable both from the angle of highlighting the issue also for European context, analysing the related time

trends (some hope for decreasing trends), high risk age groups and the periods related to childbirth and postpartum in particular.

Response: Thank you for the positive comments and recognitions for our work.

Reviewer #3 (Remarks to the Author):

<Key results: Please summarise what you consider to be the outstanding features of the work.>

This work reports the incident rate of suicide attempt is lower for mothers than fathers during the periods i) pregnancy and ii) the first 12 months of the antepartum period. This is a reversal of the relative trends seen in the 12 month period leading up to conception, where fathers display a higher incident rate than mothers.

<Validity: Does the manuscript have flaws which should prohibit its publication? If so, please provide details.>

I have some suggestions for modest improvements to be made before publication

<Originality and significance: >

My expertise is primarily in statistical methods rather than the subject matter, so I refrain from making comment on this question.

<Data & methodology: Please comment on the validity of the approach, quality of the data and quality of presentation. Please note that we expect our reviewers to review all data, including any extended data and supplementary information. Is the reporting of data and methodology sufficiently detailed and transparent to enable reproducing the results?>

I consider the approach sound and believe the data to be of high quality. Some minor comments regarding the presentation are as follows:

Response: Thank you for the positive comments and the constructive suggestions, and please find our point-by-point response below.

a) please expand on the method of Imputation of delivery date in the eMethods to clarify the process used. The sentence beginning If the length... on line 35, ending ...independent database on line 37 does not explain why the median length of stay was added to the date of admission to determine delivery date. I understand this to imply that imputed delivery dates are assumed to occur at the end of the stay.

Response: Thank you for the suggestion. For those who delivered in the hospitals, the delivery took place during the mother's hospital stay, thus the imputed delivery date must have come on or after the admission date. Most women had a very short stay in hospital for delivery and we assumed the imputed delivery date as the middle of the hospitalization; as a result, some women might have had a delivery date slightly earlier whereas others slightly later than the actual delivery date. Based on an independent database derived from Pregnancy Registry, the current imputation strategy has good validity, e.g., "97% of the pregnancies had imputed delivery date within ± 1 day within the true delivery date" (eMethods in supplementary file). We have now clarified this in the Supplementary material.

Modified text:

Supplementary file: If the length of stay was greater or equal to 3 days, the date of delivery was imputed based on parity and mode of delivery by using the date of admission plus the median length of stay from the independent database, since the delivery date must come on or after the admission date, and we assumed that the delivery happened in the middle of the stay.

b) The totals of PYs in Table 1 are consistent with the breakdowns by characteristic categories in Table 1, but they are NOT consistent with the total of PYs listed in Table S2. The IRs quoted in the first paragraph of the results use the quantities listed in Table S2. Please reconcile.

Response: Thank you for raising this issue. We checked the analytic codes and found that, in Table 1, we first grouped number of person-days by levels of covariates, and then converted person-days to person-years, by rounding off the quotient of person-days divided by 365.25 to an integer. However, in Table S2, the rounding-off of follow-up time was performed before grouping. Now, we applied the grouping-conversion approach to Table S2. The results changed minimally. We also added a footnote to Tables S2-S4 to explain the inconsistency of total person-years between Tables.

Modified text:

Table S2-S4, table note: This table was performed by collapsing data by period and sex, summing up the person-days, and then converting them to person-years. The total number of person-years in this table was not consistent with that of Table 1 because of rounding of decimals.

c) Please state the smoothing bandwidth parameters used in in Figure1, Figure 2 and Supplementary Figures S2-S6.

Response: We have now clarified in the Methods and Figure note that we used the default bandwidth in Stata, which is 0.8.

Modified text:

Page 15: In the analysis of the secular trend in risk of suicide attempt... Locally Weighted Scatterplot Smoothing, with 0.8 as the bandwidth, was used to smooth the SIR and its 95% confidence interval⁶¹.

Page 16: In the analysis of sex differences in suicide attempt... Locally Weighted Scatterplot Smoothing, with 0.8 as the bandwidth, was used to smooth the SIR and its 95% confidence interval.

d) Please elaborate on the method of standardisation by age group and calendar period referenced in the figures. In the methods section only distribution by age group was described.

Response: Thank you for the suggestion. We have now clarified it in the manuscript.

Modified text:

Page 16: *As pregnancy length is measured by week in clinical practice, in the analysis of sex differences in suicide attempt, we first calculated sex-specific incidence rates of suicide attempt by week in the preconception, antepartum and postpartum years. We then estimated SIR through direct standardization, using the distribution of age group and calendar period of the accumulated person-years during follow-up between 2001 and 2021 as the standard.*

e) Please indicate what the shaded areas are in each graph. – E.g. xx% confidence bounds and, in the methods how they were calculated

Response: The point is well taken. We have now indicated in the figure note that the shaded areas represent 95% confidence intervals for all figures in manuscript and supplementary materials. We have also clarified in the methods how we calculated them.

Modified text:

Page 15: *In the analysis of the secular trend in risk of suicide attempt, we first calculated annual sex-specific incidence rates of suicide attempt during preconception, antepartum and postpartum years, as the number of first suicide attempt divided by accumulated person-years. Then we calculated standardized incidence rates (SIR) through direct standardization, using the distribution of age group of the accumulated person-years during follow-up between 2001 and 2021 as the standard. The 95% confidence interval for SIR was calculated based on standard error⁶⁰ and two-sided Z value ($SIR \pm 1.96 * \text{standard error}$). Locally Weighted Scatterplot Smoothing, with 0.8 as the bandwidth, was used to smooth the SIR and its 95% confidence intervals⁶¹.*

Page 16: *As pregnancy length is measured by week in clinical practice, in the analysis of sex differences in suicide attempt, since pregnancy length was measured by week in clinical practice, we first calculated sex-specific incidence rates of suicide attempt by week in the preconception, antepartum and postpartum years. We then estimated SIR through direct standardization, using the distribution of age group and calendar period of the accumulated person-years during follow-up between 2001 and 2021 as the standard. The 95% confidence interval for SIR was calculated based on standard error⁶⁰ and two-sided Z value ($SIR \pm 1.96 * \text{standard error}$). Locally Weighted Scatterplot Smoothing, with 0.8 as the bandwidth, was used to smooth the SIR and its 95% confidence interval.*

Page 16: *To compare the week-specific incidence rates of suicide attempt in the antepartum and postpartum years with corresponding weeks in the preconception year, we used multivariable Poisson regression to estimate the incidence rate ratio (IRR) of maternal and paternal suicide attempt. Comparing weekly rates between antepartum and preconception period can inherently control for the seasonal variation of suicide⁶². IRRs were adjusted for country of birth, age, calendar year, education level, civil status, category of income, primiparity, history of psychiatric disorders and history of suicide attempt, all derived at the start of each period. The parameterization can be found in eMethods in Supplementary files. To investigate a potential nonlinear relationship between week and IRRs, we applied restricted cubic spline on week, and placed 4 knots at 5%, 35%, 65% and 95% on distribution of week⁶³, to visualize weekly IRRs. 95% confidence interval was calculated based on standard error estimated from delta method and z critical value.*

Page 16: *Similarly, to illustrate sex differences in the risk of suicide attempt in each period, we used multivariable Poisson regression to estimate IRR of suicide attempt, comparing mothers to fathers, weekly from preconception throughout postpartum period. To shed light*

on absolute sex differences, we further used Poisson regression to estimate the weekly incidence rate difference (IRD)⁶⁴. We also applied restricted cubic splines with 4 knots on week to visualize IRRs and IRDs. The parameterization can be found in eMethods in Supplementary files. 95% confidence interval was calculated based on standard error estimated from delta method and z value. The models were adjusted for country of birth, age, calendar year, education level, civil status, category of income, parity, history of psychiatric disorders and history of suicide attempt, all derived at the start of each period.

f) The horizontal reference line in Figure 3 would be better coloured black. The current colour of blue leads the reader to associate it exclusively with fathers.

Response: Thank you for the suggestion. The color of the reference line in Figure 3 is now black.

g) If possible, please standardise the y axis scales across preconception, antepartum and postpartum graphs in Supplementary figures S2, S7, S9

Response: Thank you for this suggestion. We have now standardized the y axis for Figure S2, S7 and S9.

h) In reference to knots used in restricted cubic splines, I assume “5%, 35%, 65% and 95% of distribution of events” corresponds to 5%, 35%, 65% and 95% of the number of weeks on the relevant x axes. Please relabel as similar, to aid in reader comprehension in Figure 3, Figure 4 and Supplementary Figures S7-S9.

Response: Thank you. We have now relabeled these figures.

Modified text:

Figure note for figure 3,4 and Supplementary figure S7-S9: A restricted cubic spline with 4 knots, placed at 5th percentile, 35th percentile, 65th percentile and 95th percentile of distribution of weeks on the relevant x axes, was used to estimate the incidence rate ratios.

i) Please state adjustment variables used in estimating IRR (currently stated for IRDs only) in Statistical analysis in methods.

Response: Thank you. We now specify in the Statistical analyses section what our models were adjusted for when estimating IRR and IRD comparing mothers to fathers; please see on page 17: “*The models were adjusted for country of birth, age, calendar year, education level, civil status, category of income, parity, history of psychiatric disorders and history of suicide attempt, all derived at the start of each period*”. We have also clarified what we adjusted for when estimating IRR during and after pregnancy compared to the corresponding week before pregnancy.

Modified text:

Page 16: To compare the week-specific incidence rates of suicide attempt in the antepartum and postpartum periods with corresponding weeks in the preconception period, we used multivariable Poisson regression to estimate the incidence rate ratio (IRR) of maternal and paternal suicide attempt. Comparing weekly rates between antepartum and preconception period can inheritably control for the seasonal variation of suicide. IRRs were adjusted for

country of birth, age, calendar year, education level, civil status, category of income, primiparity, history of psychiatric disorders and history of suicide attempt, all derived at the start of each period.

j) Please expand on the parameterisation of the assumed Poisson regression models for IRR and IRD and report parameter results. (In supplementary materials would be ok). It is apparent that the results for IRRs given in Supplementary tables S3 and S4 tabulate the cubic spline smoothed estimates.

Response: Thank you. We have now added the parameterisation and parameter results to supplementary files.

Modified text:

Page 16: To compare the week-specific incidence rates of suicide attempt in the antepartum and postpartum periods with corresponding weeks in the preconception period, we used multivariable Poisson regression to estimate the incidence rate ratio (IRR) of maternal and paternal suicide attempt. Comparing weekly rates between antepartum and preconception period can inheritably control for the seasonal variation of suicide. IRRs were adjusted for country of birth, age, calendar year, education level, civil status, category of income, primiparity, history of psychiatric disorders and history of suicide attempt, all derived at the start of each period. The parameterization can be found in eMethods in Supplementary files.

Page 16: Similarly, to illustrate sex differences in the risk of suicide attempt in each period, we used multivariable Poisson regression to estimate IRR of suicide attempt, comparing mothers to fathers, weekly from preconception throughout postpartum period. To shed light on absolute sex differences, we further used Poisson regression to estimate the weekly incidence rate difference (IRD)²⁷. We also applied restricted cubic splines with 4 knots on week to visualize IRRs and IRDs. The parameterization can be found in eMethods in Supplementary files.

Page 6: Compared to preconception period, there was a decreased risk of suicide attempt throughout the antepartum and postpartum periods among the mothers, with the lowest risk observed around delivery, e.g., at first week postpartum, Wald(1)=272.71, IRR=0.15 (95%CI: 0.12-0.18), P<0.001 at first week postpartum (Figure 3; Table S3; parameter results can be found in eMethods in Supplementary files).

Page 6: Compared to expectant fathers, expectant mothers had a similar risk of suicide attempt throughout the preconception period after adjusting for factors potentially associated with suicide attempt, e.g., history of psychiatric disorders and suicide attempt, whereas a decreased risk during antepartum and postpartum periods, with the lowest IRR noted during the first week postpartum (Wald(1)=157.89, IRR=0.22, 95%CI: 0.18-0.28, P<0.001) (Figure 4; Table S4; parameter results can be found in eMethods in Supplementary files).

Modified text:

eMethods in Supplementary files:

Poisson regression

Variables in the dataset:

Variable name	Description	Level
prd	Period	0, preconception; 1, antepartum; 2, postpartum

mot	Mother indicator	0, father; 1, mother
wksp1	Spline on week	
wksp2	Spline on week	
wksp3	Spline on week	
agec	Age group	0, 11-19; 1, 20-24; 2, 25-29; 3, 30-34; 4, 35-39; 5, ≥ 40
cal	Calendar year, group	0, 2001-2005; 1, 2006-2010; 2, 2011-2015; 3, 2016-2021
edu	Educational level (years)	1, < 10; 2, 10-12; 3, ≥ 13 ; 9, unknown
marr	Civil status	1, cohabitating; 2, non-cohabitating
income	Income level	0, Quantile 1; 1, Quantile 2; 2, Quantile 3; 3, Quantile 4; 4, Quantile 5; 9, Unknown
firstpreg	Primiparous	0, No; 1, Yes
fland	Country of birth	1, Sweden; 2, Europe; 3, Other; 9, Unknown
psy	History of psychiatric disorders	0, No; 1, Depressive disorders; 2, Other psychiatric disorders
sui_his	History of suicide attempt	0, No; 1, Yes

1) In the analysis of IRRs during and after pregnancy when compared with the corresponding week before pregnancy, the following Poisson regression was modelled among mothers and fathers separately.

$$\ln(\lambda) = \beta_0 + \beta_1 * 1.prd + \beta_2 * 2.prd + \beta_3 * wksp1 + \beta_4 * wksp2 + \beta_5 * wksp3 + \beta_6 * 1.prd * wksp1 + \beta_7 * 2.prd * wksp1 + \beta_8 * 1.prd * wksp2 + \beta_9 * 2.prd * wksp2 + \beta_{10} * 1.prd * wksp3 + \beta_{11} * 2.prd * wksp3 + \beta_{12} * 1.agec + \beta_{13} * 2.agec + \beta_{14} * 3.agec + \beta_{15} * 4.agec + \beta_{16} * 5.agec + \beta_{17} * 1.cal + \beta_{18} * 2.cal + \beta_{19} * 3.cal + \beta_{20} * 2.edu + \beta_{21} * 3.edu + \beta_{22} * 9.edu + \beta_{23} * 2.marr + \beta_{24} * 1.income + \beta_{25} * 2.income + \beta_{26} * 3.income + \beta_{27} * 4.income + \beta_{28} * 9.income + \beta_{29} * 1.firstpreg + \beta_{30} * 2.fland + \beta_{31} * 3.fland + \beta_{32} * 9.fland + \beta_{33} * 1.psy + \beta_{34} * 2.psy + \beta_{35} * 1.sui_his$$

Parameter results among mothers:

_d	IRR	Std. err.	z	P> z	[95% conf. interval]	

-----+						

prd						
1	.9543514	.0809318	-0.55	0.582	.8082094	1.126919
2	.1347109	.0171179	-15.78	0.000	.1050121	.1728088
wksp1	.9782664	.0050541	-4.25	0.000	.9684105	.9882227
wksp2	1.067912	.0199297	3.52	0.000	1.029556	1.107696
wksp3	.836969	.042856	-3.48	0.001	.7570502	.9253246
prd#c.wksp1						
1	.9083492	.0095288	-9.16	0.000	.8898637	.9272187
2	1.082653	.0127071	6.77	0.000	1.058032	1.107847

```

|
prd#c.wksp2 |
  1 | 1.30783 .0610098 5.75 0.000 1.193556 1.433044
  2 | .8510833 .0325917 -4.21 0.000 .7895431 .9174201
|
prd#c.wksp3 |
  1 | .4835254 .0766143 -4.59 0.000 .3544434 .6596169
  2 | 1.44629 .146515 3.64 0.000 1.185837 1.763947
|
agec |
20- | .6145662 .0253528 -11.80 0.000 .5668314 .6663209
25- | .3926078 .0179395 -20.46 0.000 .3589756 .429391
30- | .3513518 .017752 -20.70 0.000 .3182257 .3879262
35- | .3291343 .0193346 -18.92 0.000 .2933393 .3692972
40- | .3061849 .0282587 -12.82 0.000 .2555195 .3668965
|
cal |
2006- | .9262474 .0301603 -2.35 0.019 .8689811 .9872877
2011- | .7785892 .0273933 -7.11 0.000 .7267087 .8341736
2016- | .7047874 .0271054 -9.10 0.000 .6536146 .7599666
|
edu |
  2 | .6596962 .0198587 -13.82 0.000 .6218998 .6997897
  3 | .3993398 .0160692 -22.81 0.000 .3690548 .4321101
  9 | .9308803 .0581456 -1.15 0.252 .8236168 1.052113
|
2.marr | 1.979317 .0569594 23.73 0.000 1.870769 2.094164
|
income |
  1 | .8930386 .0277571 -3.64 0.000 .8402597 .9491328
  2 | .7413197 .0280904 -7.90 0.000 .6882583 .7984718
  3 | .5994064 .027448 -11.18 0.000 .5479528 .6556915
  4 | .5897703 .0321417 -9.69 0.000 .5300215 .6562545
  9 | .9683281 .1231623 -0.25 0.800 .754671 1.242474
|
1.firstpreg | 1.094256 .0312532 3.15 0.002 1.034684 1.157258
|
fland |
  2 | 1.033588 .0492681 0.69 0.488 .9413975 1.134806
  3 | 1.071931 .0382499 1.95 0.052 .9995243 1.149584
  9 | 2.966497 2.967741 1.09 0.277 .4175277 21.07669
|
psy |
  1 | 3.413934 .1227756 34.14 0.000 3.181583 3.663253
  2 | 3.391304 .1002154 41.33 0.000 3.200466 3.593523
|
1.sui_his | 5.461194 .1553132 59.69 0.000 5.165114 5.774246
_cons | .0000169 1.21e-06 -154.18 0.000 .0000147 .0000195
ln(fu) | 1 (exposure)
-----

```

Parameter results among fathers:

_d	IRR	Std. err.	z	P> z	[95% conf. interval]	
prd						
1	1.220455	.10921	2.23	0.026	1.024127	1.45442
2	.701637	.0659669	-3.77	0.000	.5835576	.8436091
wksp1						
	.9873805	.0062333	-2.01	0.044	.9752388	.9996733
wksp2						
	1.031417	.0234201	1.36	0.173	.9865213	1.078357
wksp3						
	.9157926	.0570516	-1.41	0.158	.8105307	1.034725
prd#c.wksp1						
1	.988193	.0094686	-1.24	0.215	.9698081	1.006926
2	1.034034	.0096659	3.58	0.000	1.015261	1.053153
prd#c.wksp2						
1	1.054009	.0402702	1.38	0.169	.9779641	1.135968
2	.9155742	.029895	-2.70	0.007	.8588166	.9760828
prd#c.wksp3						
1	.8164176	.0990784	-1.67	0.095	.6435951	1.035648
2	1.263625	.1119092	2.64	0.008	1.062268	1.50315
agec						
20-	.8983597	.0643455	-1.50	0.135	.780697	1.033756
25-	.6556727	.0467538	-5.92	0.000	.5701521	.754021
30-	.5344667	.0389579	-8.59	0.000	.4633142	.6165464
35-	.4639209	.0351285	-10.14	0.000	.3999357	.5381429
40-	.4072848	.031982	-11.44	0.000	.3491867	.4750493
cal						
2006-	1.089244	.0341098	2.73	0.006	1.0244	1.158192
2011-	.9481036	.0319374	-1.58	0.114	.8875292	1.012812
2016-	.8356237	.0307219	-4.88	0.000	.7775282	.89806
edu						
2	.7136822	.0197446	-12.19	0.000	.6760139	.7534494
3	.4264992	.0160126	-22.70	0.000	.396242	.4590669
9	.8497429	.074097	-1.87	0.062	.7162479	1.008119
2.marr						
	2.204101	.0609242	28.59	0.000	2.087869	2.326804
income						
1	.8870722	.0277042	-3.84	0.000	.8344015	.9430677
2	.6824647	.0238973	-10.91	0.000	.637198	.7309472
3	.6249241	.0234478	-12.53	0.000	.5806163	.672613
4	.5403733	.0229227	-14.51	0.000	.4972627	.5872215
9	.388835	.0658098	-5.58	0.000	.2790619	.5417889

1.firstpreg		.9398038	.0228912	-2.55	0.011	.8959921	.9857579
fland							
2		.8812454	.0382634	-2.91	0.004	.8093529	.9595239
3		.6910945	.0263446	-9.69	0.000	.6413418	.7447069
9		9.78e-06	.0042401	-0.03	0.979	0	.
psy							
1		2.574598	.1017255	23.93	0.000	2.382744	2.781899
2		2.756774	.0752827	37.13	0.000	2.613102	2.908345
1.sui_his							
_cons		.0000106	1.04e-06	-117.33	0.000	8.78e-06	.0000129
ln(fu)		1 (exposure)					

2) In the analysis of IRRs and IRDs of suicide attempt comparing mothers to fathers, the following Poisson regression was modelled among preconception, antepartum and postpartum period separately.

$$\ln(\lambda) = \beta_0 + \beta_1 * 1.mot + \beta_2 * wksp1 + \beta_3 * wksp2 + \beta_4 * wksp3 + \beta_5 * 1.mot * wksp1 + \beta_6 * 1.mot * wksp2 + \beta_7 * 1.mot * wksp3 + \beta_8 * 1.agec + \beta_9 * 2.agec + \beta_{10} * 3.agec + \beta_{11} * 4.agec + \beta_{12} * 5.agec + \beta_{13} * 1.cal + \beta_{14} * 2.cal + \beta_{15} * 3.cal + \beta_{16} * 2.edu + \beta_{17} * 3.edu + \beta_{18} * 9.edu + \beta_{19} * 2.marr + \beta_{20} * 1.income + \beta_{21} * 2.income + \beta_{22} * 3.income + \beta_{23} * 4.income + \beta_{24} * 9.income + \beta_{25} * 1.firstpreg + \beta_{26} * 2.fland + \beta_{27} * 3.fland + \beta_{28} * 9.fland + \beta_{29} * 1.psy + \beta_{30} * 2.psy + \beta_{31} * 1.sui_his$$

Parameter results in the preconception period:

_d	IRR	Std. err.	z	P> z	[95% conf. interval]		
1.mot		1.064456	.082743	0.80	0.422	.9140324	1.239635
wksp1		.9875772	.0060411	-2.04	0.041	.9758075	.9994888
wksp2		1.030843	.022688	1.38	0.168	.9873209	1.076284
wksp3		.9211068	.052507	-1.44	0.149	.8237357	1.029988
mot#c.wksp1							
1		.9908787	.0078317	-1.16	0.246	.9756471	1.006348
mot#c.wksp2							
1		1.034364	.029464	1.19	0.236	.9781975	1.093754
mot#c.wksp3							
1		.9210479	.0679529	-1.11	0.265	.7970442	1.064344
agec							
20-		.6370061	.0266227	-10.79	0.000	.5869064	.6913824
25-		.4105981	.0185714	-19.68	0.000	.3757655	.4486595
30-		.3443996	.0172164	-21.32	0.000	.3122565	.3798514
35-		.3125632	.0181815	-19.99	0.000	.2788845	.350309

```

40- | .2660655 .02008 -17.54 0.000 .2294819 .3084813
|
cal |
2006- | .9719344 .031166 -0.89 0.375 .9127301 1.034979
2011- | .8911274 .0311675 -3.30 0.001 .832087 .9543571
2016- | .8307962 .0326788 -4.71 0.000 .7691535 .8973791
|
edu |
2 | .6475401 .0189087 -14.88 0.000 .6115203 .6856815
3 | .3382274 .0139264 -26.33 0.000 .3120045 .3666542
9 | .8291675 .057793 -2.69 0.007 .7232916 .9505415
|
2.marr | 1.971255 .057867 23.12 0.000 1.861038 2.087998
|
income |
1 | .8528983 .0271965 -4.99 0.000 .8012257 .9079035
2 | .6889373 .0254394 -10.09 0.000 .6408385 .7406463
3 | .6500324 .0260862 -10.73 0.000 .6008634 .703225
4 | .5296118 .0253361 -13.29 0.000 .4822108 .5816722
9 | .707896 .0910932 -2.68 0.007 .5500922 .9109686
|
1.firstpreg | 1.068607 .0308317 2.30 0.021 1.009855 1.130777
|
fland |
2 | .9318005 .0453724 -1.45 0.147 .8469839 1.025111
3 | .8949121 .0342484 -2.90 0.004 .8302423 .9646193
9 | 2.83623 2.837741 1.04 0.297 .3991043 20.15564
|
psy |
1 | 3.313666 .1233692 32.18 0.000 3.080478 3.564506
2 | 3.21406 .0933811 40.19 0.000 3.03615 3.402395
|
1.sui_his | 5.803408 .167741 60.84 0.000 5.483781 6.141665
_cons | .0000165 1.32e-06 -137.01 0.000 .0000141 .0000193
ln(fu) | 1 (exposure)

```

Parameter results in the antepartum period:

_d	IRR	Std. err.	z	P> z	[95% conf. interval]
1.mot	.9840788	.0994798	-0.16	0.874	.8072026 1.199713
wksp1	.9709463	.0088358	-3.24	0.001	.9537819 .9884195
wksp2	1.078648	.0320373	2.55	0.011	1.017649 1.143303
wksp3	.8049219	.0689392	-2.53	0.011	.6805356 .9520432
mot#c.wksp1					
1	.8899837	.0129138	-8.03	0.000	.8650296 .9156578
mot#c.wksp2					
1	1.291672	.064895	5.09	0.000	1.170542 1.425337

```

      |
mot#c.wksp3 |
      1 | .5485086 .0810658 -4.06 0.000 .4105647 .7327996
      |
      agec |
      20- | .8552922 .08247 -1.62 0.105 .7080092 1.033213
      25- | .6517477 .0637065 -4.38 0.000 .5381172 .7893727
      30- | .5379443 .0544076 -6.13 0.000 .4412117 .6558849
      35- | .5031529 .0534871 -6.46 0.000 .4085205 .6197065
      40- | .4120265 .0478978 -7.63 0.000 .3280754 .5174599
      |
      cal |
      2006- | 1.146776 .0571439 2.75 0.006 1.040072 1.264428
      2011- | .9517903 .0504883 -0.93 0.352 .8578054 1.056073
      2016- | .793157 .0456438 -4.03 0.000 .7085576 .8878575
      |
      edu |
      2 | .7275324 .0327755 -7.06 0.000 .666048 .7946926
      3 | .4717185 .0273582 -12.96 0.000 .4210328 .5285061
      9 | .9926545 .1128715 -0.06 0.948 .794348 1.240468
      |
      2.marr | 2.07558 .0985344 15.38 0.000 1.891169 2.277974
      |
      income |
      1 | .8994686 .0449014 -2.12 0.034 .8156317 .9919229
      2 | .7246813 .0404782 -5.77 0.000 .649534 .8085227
      3 | .5965922 .0363429 -8.48 0.000 .5294494 .6722499
      4 | .5964382 .0389205 -7.92 0.000 .524832 .677814
      9 | .3742848 .0973557 -3.78 0.000 .2247994 .623174
      |
      1.firstpreg | 1.113567 .0424497 2.82 0.005 1.033399 1.199954
      |
      fland |
      2 | .9016069 .0622097 -1.50 0.133 .7875632 1.032165
      3 | .8241583 .0463477 -3.44 0.001 .7381457 .9201936
      9 | .000022 .0097405 -0.02 0.981 0 .
      |
      psy |
      1 | 2.489842 .1437397 15.80 0.000 2.223472 2.788124
      2 | 2.463049 .107715 20.61 0.000 2.260727 2.683479
      |
      1.sui_his | 6.777572 .3019819 42.95 0.000 6.210806 7.396058
      _cons | .0000117 1.47e-06 -90.38 0.000 9.11e-06 .0000149
      ln(fu) | 1 (exposure)

```

Parameter results in the postpartum period:

_d	IRR	Std. err.	z	P> z	[95% conf. interval]
1.mot	.2166467	.0279135	-11.87	0.000	.1682986 .2788841

wksp1		1.01754	.0061804	2.86	0.004	1.005499	1.029726
wksp2		.9592095	.0177426	-2.25	0.024	.9250574	.9946224
wksp3		1.121091	.060435	2.12	0.034	1.008683	1.246025
mot#c.wksp1							
1		1.034857	.0113808	3.12	0.002	1.012789	1.057405
mot#c.wksp2							
1		.9679229	.0308813	-1.02	0.307	.9092502	1.030382
mot#c.wksp3							
1		1.036821	.094569	0.40	0.692	.8670921	1.239773
agec							
20-		.7683854	.0710241	-2.85	0.004	.641062	.9209968
25-		.5581793	.0518766	-6.27	0.000	.4652259	.669705
30-		.4849325	.0460429	-7.62	0.000	.4025893	.5841175
35-		.4055881	.0400922	-9.13	0.000	.3341522	.4922959
40-		.3681947	.0382653	-9.61	0.000	.3003411	.4513779
cal							
2006-		.9698459	.0400135	-0.74	0.458	.894508	1.051529
2011-		.7461653	.0330355	-6.61	0.000	.6841466	.813806
2016-		.6705899	.0312142	-8.58	0.000	.612119	.734646
edu							
2		.7355279	.0269081	-8.40	0.000	.6846354	.7902035
3		.4914718	.0235368	-14.83	0.000	.4474394	.5398374
9		1.092079	.1048687	0.92	0.359	.9047239	1.318233
2.marr		2.367436	.0825769	24.71	0.000	2.210997	2.534944
income							
1		.9443121	.0361291	-1.50	0.134	.8760902	1.017847
2		.759725	.0349896	-5.97	0.000	.6941507	.8314939
3		.6185611	.0344333	-8.63	0.000	.5546244	.6898684
4		.6186159	.0406772	-7.30	0.000	.5438138	.7037073
9		.4602934	.102074	-3.50	0.000	.2980393	.7108794
1.firstpreg		.8919165	.0281654	-3.62	0.000	.8383871	.9488638
fland							
2		.9787987	.0532841	-0.39	0.694	.8797422	1.089009
3		.8271065	.0375779	-4.18	0.000	.7566392	.9041366
9		8.68e-06	.0052365	-0.02	0.985	0	.
psy							
1		2.675431	.1299931	20.25	0.000	2.432404	2.942738
2		3.087397	.1096126	31.75	0.000	2.879865	3.309885

l.sui_his	5.829602	.2149795	47.81	0.000	5.423117	6.266555
_cons	8.73e-06	1.03e-06	-98.53	0.000	6.92e-06	.000011
ln(fu)	1 (exposure)					

No pre-registration

Statistical methods are appropriate conditional on concern a) being addressed.

a) I do not understand why the preconception incident rate (“baseline”) is calculated for each of 52 weeks. This implies there is specific interest in the way incident rate during this period changes over time (e.g. see b) below), and that the relationship between the incident rate of a given week of either pregnancy or postpartum is related to the incident rate of the exact corresponding week in preconception.

If so, this needs to be explained in the methods.

If not, I believe it would be better (and remove noise associated with random fluctuations in the preconception period) to compare each week of pregnancy and the postpartum period to a single baseline estimate for each of mothers and fathers from the preconception period. There could be several valid estimates of such baseline estimates; any reasonable well defined method would suffice.

Response: Thank you for this valid point. In clinical practice, pregnancy length was measured in weeks so we believe calculating and comparing incidence rates in week can be informative to clinicians. In addition, comparing antepartum to preconception period by week can inheritably control for potential seasonal variation in the risk of suicide^[1]. We have clarified it in the manuscript.

Modified text:

Page 16: As pregnancy length is measured by week in clinical practice, in the analysis of sex differences in suicide attempt, we first calculated sex-specific incidence rates of suicide attempt by week in the preconception, antepartum and postpartum years.

Page 16: To compare the week-specific incidence rates of suicide attempt in the antepartum and postpartum periods with corresponding weeks in the preconception period, we used multivariable Poisson regression to estimate the incidence rate ratio (IRR) of maternal and paternal suicide attempt. Comparing weekly rates between antepartum and preconception period can inheritably control for the seasonal variation of suicide⁶².

However, we agree with the later comment from the reviewer that “the variation introduced by comparing antepartum and postpartum rates to fluctuating preconception rates” may make the IRR hard to explain to some extent; thus, we estimated IRRs by using the average rate of suicide attempt in the preconception year as the reference (please see **Figure S10** below). The pattern of IRR is very similar to that generated by weekly comparison.

Modified text:

Page 17: In addition, since the SIR in preconception period fluctuated over weeks, we also estimated IRR comparing the antepartum and the postpartum to the preconception period, by using the average incidence rate in the preconception year as the reference.

Page 7: IRR calculated by comparing antepartum and postpartum periods to average incidence rate of preconception period yielded similar results on trends (Figure S10).

Figure S10 Incidence rate ratio of parental suicide attempt during and after pregnancy when compared with average incidence rate before pregnancy

The incidence rate ratios were adjusted for country of birth, age, calendar year, education level, civil status, category of income, primiparity, history of psychiatric disorders and history of suicide attempt, all derived at the start of each period. The shaded areas indicate 95% confidence interval.

In addition, we rephrased our interpretation regarding paternal suicide attempt, by clarifying that the rising paternal risk in the later postpartum period was observed only when compared with the same week in the preconception period.

Modified text:

Page 3, Abstract: This reversed sex difference in parental risk of suicide attempt during and after pregnancy compared to that in the general population may suggest that pregnancy or childbirth has a more pronounced effect among mothers than fathers.

Page 7, first paragraph of Discussion: Compared to the year before pregnancy, mothers had a decreased risk of suicide attempt throughout pregnancy and one year postpartum.

However, fathers had a decreased risk around childbirth, and a higher risk of suicide attempt in the later postpartum period only when compared to corresponding weeks in preconception year.

Page 11, “Sex difference in postpartum suicide attempt”, Discussion: When compared to the preconception period, mothers still had a decreased risk of suicide attempt throughout the postpartum period, whereas fathers only had a decreased risk in the first half of the year postpartum. Such difference may be related to that mothers are usually the primary caregivers during the first months postpartum and may therefore develop closer emotional bonds with their children than fathers, which may in turn buffer poor mental health in mothers⁴¹. In addition, in some qualitative studies, fathers claimed that difficulties in

balancing needs of family and work were important sources of stress in the year after birth⁵¹. In addition, fathers may experience hormonal and neurofunctional alterations after childbirth⁵², but more research is needed to understand whether these biological changes are associated with paternal suicide attempt postpartum. Since mental health screening is not in place for fathers yet, health providers may stay alert of paternal risk during the postpartum period.

Page 12, last paragraph of Discussion: Clinicians may take opportunities of the current scheme of maternal and child health services to identify high-risk individuals, implement suicide risk assessment, and enhance suicide prevention strategies for both mothers and fathers.

Reference:

1. To S, Messias E, Burch L, Chibnall J. Seasonal variation in suicide: age group and summer effects in the United States (2015–2020). *BMC Psychiatry*. 2024;24:856. doi:10.1186/s12888-024-06309-7

b) It is interesting to note the apparent decline in incident rates (and standardised incident rates) for both mothers and fathers during the 12 month leadup to conception. Is there an explanation for this?

Response: We would like to clarify that the curves were smoothed by LOWESS, and there is no obvious trend of the scattered dots from visual inspection. We also formally tested the magnitude of the trend by week using Poisson regression. When analyzing the association between week and risk of suicide attempt in the preconception year, we found that the incidence rate remained largely unchanged over weeks (**Table R1**). Although the effect sizes are statistically significant, it is possibly due to our large sample size, which makes it oversensitive to minor deviation from the null hypothesis. We also performed the piecewise Poisson regressions to detect potential non-linear associations, but these also yielded very stable incidence rates across different periods in the preconception year (**Table R1**). In addition, we also did linear regression, by using the age and calendar year-standardized incidence rates per 1000 person years as dependent variable, and week (continuous variable) as independent variable. The results also showed no statistically significant trend.

Table R1 Association between week and incidence rate of suicide attempt in the preconception year

	Mothers		Fathers	
	IRR (95% CI)	P-value	IRR (95% CI)	P-value
Poisson regression	1.00 (0.99-1.00)*	<0.001	1.00 (0.99-1.00)*	<0.001
Piecewise Poisson regression ^a				
week ≤15	0.98 (0.97-0.99)*	0.005	0.99 (0.98-1.00)	0.182
week >15 and ≤35	1.00 (1.00-1.01)	0.249	1.00 (0.99-1.01)	0.793
week >35	0.99 (0.98-1.00)*	0.031	0.99 (0.98-1.00)	0.117
Linear regression	-0.01 (-0.01-0.00)	0.051	-0.01 (-0.02--0.00)*	<0.001

CI, confidence interval; IRR, incidence rate ratio.

Poisson regression was adjusted for week (continuous variable), and groups of age and calendar period. Linear regression was performed by using age and calendar year-standardized incidence rates per 1000 person years as dependent variable, and week (continuous variable) as independent variable.

^aThe cutoff was chosen based on visual inspection of turning points indicated in Figure 2.

We have also discussed the trend in preconception period in the manuscript.

Modified text:

Page 9: Sex difference in preconceptional suicide attempt

Preconception is considered as a relative healthy period when expectant mothers plan for pregnancy³³, and possibly for fathers as well. Our study observed a largely stable, given the wide CIs, SIR of suicide attempt over the weeks in preconception period, compared to the trend during pregnancy. This suggests such “health bias” may not significantly affect the weekly trend of suicide attempt before pregnancy.

Is there a similar pattern in the 12 months prior to the beginning of the of the 12 month leadup to conception (ie beginning 2 years before conception)?

Response: Thank you for this interesting question. We estimated the standardized incidence rate of parental suicide attempt in two years before conception (please see **Figure R2** below). The statistical test showed no pronounced trend in either mothers or fathers over weeks in relation to two years before conception (**Table R2**). Due to limited space, we did not include this figure in the manuscript.

Figure R2 Standardized incidence rate of parental suicide attempt two years before conception

Incidence rates were standardized by distribution of age group and calendar period of the accumulated person-years during follow-up. Locally Weighted Scatterplot Smoothing (bandwidth=0.8) was used to estimate the trend (solid line), while the dots indicate the weekly incidence rates. The shaded areas indicate 95% confidence interval.

Table R2 Association between week and incidence rate of suicide attempt two years before conception

	Mothers	Fathers
--	---------	---------

	IRR (95% CI)	P value	IRR (95% CI)	P value
Poisson regression	1.00 (1.00-1.00)*	0.012	1.00 (0.99-1.00)*	0.009
Piecewise Poisson regression ^a				
week ≤30	0.99 (0.99-1.00)*	0.027	0.99 (0.99-1.00)*	0.032
week > 30	1.00 (0.99-1.01)	0.544	1.00 (0.99-1.01)	0.852
Linear regression	-0.01 (-0.01-0.00)	0.078	-0.01 (-0.01--0.00)	0.036

CI, confidence interval; IRR, incidence rate ratio.

Poisson regression was adjusted for week (continuous variable), and groups of age and calendar period. Linear regression was performed by using age and calendar year-standardized incidence rates per 1,000 person years as dependent variable, and week (continuous variable) as independent variable.

^aThe cutoff was chosen based on visual inspection of turning points indicated in Figure R2.

It is also surprising to note a modest correlation (r=0.5) between the incident rates for mothers and fathers across the 53 weeks of preconception. Was this expected by the authors?

Response: Thank you for raising this point. We are not aware of studies showing a correlation between expectant parents' suicide behaviour specifically in the preconception period, but some literature has pointed out associations between suicide between spouses regardless of their life stage ^[1,2], possibly due to the stress brought by suicide of a partner, shared environmental factors between the cohabitating partners or selective mating. Therefore such association was expected and discussed in the manuscript.

Modified text:

Page 9: Sex difference in preconceptional suicide attempt

In addition, we found a similar risk of suicide attempt among mothers and fathers over weeks throughout the preconception period. The trend of weekly SIRs in preconception period among fathers somewhat echoed the trend among mothers. This is consistent with previous literature reporting a positive link on suicide risk between spouses, possibly due to stress brought by suicide of a partner, or shared environmental factors between the cohabitating partners or selective mating ^{34,35}.

References:

1. Agerbo E. Risk of suicide and spouse's psychiatric illness or suicide: nested case-control study. *BMJ*. 2003;327(7422):1025-1026.
2. Jang J, Park SY, Kim YY, et al. Risks of suicide among family members of suicide victims: A nationwide sample of South Korea. *Front Psychiatry*. 2022;13:995834. doi:10.3389/fpsy.2022.995834

<Custom code: If the work includes custom code, does the code run as intended? If you are unable to access the code, please contact us. >

No custom code. I believe all analyses would have been run with standard code accessible in the statistical packages referenced.

Response: Thank you. We did run all the analyses with standard code. The codes are now available on GitHub (<https://github.com/yihuiyang2/ParentalSuicide>).

<Conclusions:> Do you find that the conclusions and data interpretation are robust, valid and reliable?

Conclusions and data interpretation appear valid. However, when considering the raw data in Figure 2, the following interpretation may be considered: Fathers' suicide attempt incident rate is relatively constant from the beginning of the preconception period to the end of the postpartum period, with the exception of a dip during the period approximately 5-10 weeks gestation to approximately 5-10 weeks postpartum. Coinciding with this, mothers suicide attempt incident rate drops markedly from the beginning of pregnancy and begins to rise at approximately 6 weeks postpartum before stabilising at approximately 6 months postpartum to a level which remains much lower than mothers' preconception levels and fathers' levels at the end of postpartum.

Given the apparent unexplained decreasing trend for IRs in Figure 2(Preconception), the variation introduced by comparing antepartum and postpartum rates to fluctuating preconception rates, caution surrounding estimates from weeks 53 and the arbitrary nature of cubic splines, the IRR trends displayed in Figure 3 may have accumulated distortions.

Response: The point is well taken.

First, our earlier response has shown there is no statistically significant decline in SIR in the preconception period (page 23 in this letter).

Second, our analyses by comparing the incidence rates during the antepartum and postpartum periods to a single baseline rate generated similar IRRs (page 21-22 in this letter).

Third, we estimated IRRs by excluding week 53 in the analysis, and found similar results (**Figure S12**).

Figure S12 Incidence rate ratio of parental suicide attempt during and after pregnancy when compared with the corresponding week before pregnancy, after excluding week 53 in the analysis

The follow-up during pregnancy was censored at week 40 due to few subsequent events.

A restricted cubic spline with 4 knots, placed at 5th percentile, 35th percentile, 65th percentile and 95th percentile of distribution of weeks on the relevant x axis, was used to estimate the incidence rate ratios. The incidence rate ratios were adjusted for country of birth, age, calendar year, education level, civil status, category of income, primiparity, history of psychiatric disorders and history of suicide attempt, all derived at the start of each period. The shaded areas indicate 95% confidence interval.

Fourth, we also estimated IRRs without using splines and found comparable results (**Figure S14**). IRRs comparing mothers to fathers generated comparable results as well (**Figure S15**).

Figure S14 Incidence rate ratio of parental suicide attempt during and after pregnancy when compared with the corresponding week before pregnancy: a nationwide register-based study in Sweden, 2001-2021

The follow-up during pregnancy was censored at week 40 due to few subsequent events. The incidence rate ratios were adjusted for country of birth, age, calendar year, education level, civil status, category of income, primiparity, history of psychiatric disorders and history of suicide attempt, all derived at the start of each period. The shaded areas indicate 95% confidence interval.

Figure S15 Incidence rate ratio of suicide attempt among mothers when compared with the corresponding week among fathers in preconception, antepartum and postpartum period

The follow-up during pregnancy was censored at week 40 due to few subsequent events. The incidence rate ratios were adjusted for country of birth, age, calendar year, education level, civil status, category of income, primiparity, history of psychiatric disorders and history of suicide attempt, all derived at the start of each period. The shaded areas indicate 95% confidence interval.

We have included above figures in the manuscript.

Modified text:

Page 17: In addition, due to the small number of person-years, we excluded week 53 from analyses, and calculated SIR by week, and estimated IRRs comparing antepartum and postpartum to preconception, and comparing mothers to fathers in preconception and postpartum. Finally, we estimated IRRs comparing during and after pregnancy to before pregnancy, and IRRs comparing mothers to fathers, when not using the splines.

Page 7: Furthermore, when excluding week 53 from the analysis, the rise in paternal absolute IR in late postpartum period was less pronounced yet the CIs overlapped with those from the primary analysis (Figure S11) whereas the IRRs changed minimally (Figure S12-S13). In addition, pattern of IRRs comparing periods and parents when not using splines was similar to that when using splines (FigureS14-S15).

I would caution against conclusions that refer to specific time points e.g. “an increased risk from the 37th week postpartum”. Firstly, the ‘risk’ is the average risk, estimated from the combination of several modelling choices. Secondly, there is no obvious step function at 37 weeks, so I would suggest precision no more than ‘at around 37 weeks’ .

Response: We agree with the reviewer’s concern. We have now rephrased our conclusion regarding paternal suicide, by removing the statement on the exact time point and clarifying the rising paternal risk in the later postpartum period was seen only when compared with the same weeks in the preconception period.

Modified text:

Page 3, Abstract: Compared to the *corresponding week* in the preconception period, mothers had a lower risk of suicide attempt throughout pregnancy and postpartum period; fathers had a decreased risk of suicide attempt around childbirth, but *a higher risk in the later postpartum period*. Compared to fathers, mothers had a lower risk of suicide attempt during and after pregnancy. This reversed sex difference in parental risk of suicide attempt during and after pregnancy compared to that in the general population may suggest that *pregnancy or childbirth has a more pronounced effect among mothers than fathers*.

Page 6, Result: Although a similar U-shaped pattern was observed for fathers, the risk of paternal suicide attempt significantly decreased during the first 8 weeks postpartum *and became higher in the later postpartum period*, when compared to corresponding weeks in the preconception period (Figure 3).

Page 7-8, first paragraph of Discussion: Compared to the year before pregnancy, mothers had a decreased risk of suicide attempt throughout pregnancy and one year postpartum. *However, fathers had a decreased risk around childbirth, and a higher risk of suicide attempt in the later postpartum period only when compared to corresponding weeks in preconception year*.

Page 9-10, “Sex differences in antepartum suicide attempt”, Discussion: In addition, our results also indicated a decreased risk of paternal suicide attempt *towards the end of antepartum only when compared to corresponding weeks in the preconception period*, although the IRR did not reach statistical significance due to low number of cases.

Page 11, “Sex difference in postpartum suicide attempt”, Discussion: When compared to the preconception period, mothers still had a decreased risk of suicide attempt throughout the postpartum period, whereas fathers only had a decreased risk in the first half of the year postpartum. Such difference may be related to that mothers are usually the primary caregivers during the first months postpartum and may therefore develop closer emotional bonds with their children than fathers, which may in turn buffer poor mental health in mothers⁴¹. *In addition, in some qualitative studies, fathers claimed that difficulties in balancing needs of family and work were important sources of stress in the year after birth*⁵¹.

In addition, fathers may experience hormonal and neurofunctional alterations after childbirth⁵², but more research is needed to understand whether these biological changes are associated with paternal suicide attempt postpartum. Since mental health screening is not in place for fathers yet, health providers may stay alert of paternal risk during the postpartum period.

Page 12, last paragraph of Discussion: Clinicians may take opportunities of the current scheme of maternal and child health services to identify high-risk individuals, implement suicide risk assessment, and enhance suicide prevention strategies for both mothers and fathers.

I am puzzled by the results presented in Figure 4, Preconception. I agree with the conclusion from the data presented in Figure 4— that is “Compared to fathers, mothers had a similar risk of suicide attempt throughout the preconception period”. However, this is at odds with the precis of the introduction “in the general population, women are more likely than men to attempt suicide”, the general pattern observed in Figure 1 (Preconception), and the statement in the Results paragraph headed Secular trends in parental suicide attempt, namely “Compared to fathers, mothers had higher incidence rates of suicide attempt in the preconception period...”.

Response: This point is well taken.

The female-predominance in suicide attempt in the general population was mostly based on comparison of prevalence or rates^[1,2]. In Figure 1, the incidence rate was only standardized by age, without further adjustment. However, in Figure 4, we showed IRRs comparing mothers to fathers after adjustment for multiple potential risk factors for suicide attempt. Some of these factors, e.g., history of psychiatric disorders and suicide attempt, are more common among mothers than fathers, and adjusting them may dilute the female-predominance in suicide attempt. Indeed, when we analyzed crude IRRs, mothers were more likely to attempt suicide than fathers (please see **Figure R4** below). Therefore, we believe the divergence between Figure 1 and Figure 4 is due to adjustment of covariates. We have now clarified this in the manuscript.

Figure R4 Incidence rate ratio of suicide attempt among mothers when compared with the corresponding week among fathers in the preconception year

A restricted cubic spline with 4 knots, placed at 5th percentile, 35th percentile, 65th percentile and 95th percentile of distribution of weeks on the relevant x axis, was used to estimate the incidence rate ratios. The incidence rate ratios were not adjusted for any covariates (namely, country of birth, age, calendar year, education level, civil status, category of income, primiparity, history of psychiatric disorders or history of suicide attempt). The shaded areas indicate 95% confidence interval.

Modified text:

Page 6: Compared to expectant fathers, expectant mothers had a similar risk of suicide attempt throughout the preconception period, after adjusting for factors potentially associated with suicide attempt, e.g., history of psychiatric disorders and suicide attempt, whereas a decreased risk during antepartum and postpartum periods, with the lowest IRR noted during the first week postpartum (Wald(1)=157.89, IRR=0.22, 95%CI: 0.18-0.28, $P<0.001$) (Figure 4; Table S4; parameter results can be found in eMethods in Supplementary files).

Reference:

1. Turecki G, Brent DA. Suicide and suicidal behaviour. Lancet. 2016;387(10024):1227-1239. doi:10.1016/S0140-6736(15)00234-2
2. Zhang J, McKeown RE, Hussey JR, Thompson SJ, Woods JR. Gender differences in risk factors for attempted suicide among young adults: findings from the Third National Health and Nutrition Examination Survey. Ann Epidemiol. 2005;15(2):167-174. doi:10.1016/j.annepidem.2004.07.095

<Suggested improvements: Please list additional analyses, experiments or data that could help strengthening the work in a revision.>

I suggest investigating the effect/s, if any of excluding week 53 from analyses. This is due

to the much smaller number of PYs available, and the apparent ‘kick-up’ in the SIR for fathers for week 53 postpartum (Figure 2).

Response: Thank you for this helpful suggestion. We have now excluded week 53 from the analysis. Although the IRRs comparing the antepartum and postpartum to the preconception period, and those comparing mothers to fathers in the preconception and postpartum period, changed minimally (Figure S12 and S13), the rise in paternal absolute IR in the late postpartum period was no longer noted (Figure S11 below).

Figure S11 Standardized incidence rate of parental suicide attempt before, during, and after pregnancy by week, after excluding week 53 from the analysis

The follow-up during pregnancy was censored at week 40 due to few subsequent events.

Incidence rates were standardized by distribution of age group and calendar period of the accumulated person-years during follow-up. Locally Weighted Scatterplot Smoothing (bandwidth=0.8) was used to estimate the trend (solid line), while the dots indicate the weekly incidence rates. The shaded areas indicate 95% confidence interval.

Figure S12 Incidence rate ratio of parental suicide attempt during and after pregnancy when compared with the corresponding week before pregnancy, after excluding week 53 from the analysis

The follow-up during pregnancy was censored at week 40 due to few subsequent events. A restricted cubic spline with 4 knots, placed at 5th percentile, 35th percentile, 65th percentile and 95th percentile of distribution of weeks on the relevant x axis, was used to estimate the incidence rate ratios. The incidence rate ratios were adjusted for country of birth, age, calendar year, education level, civil status, category of income, primiparity, history of psychiatric disorders and history of suicide attempt, all derived at the start of each period. The shaded areas indicate 95% confidence interval.

Figure S13 Incidence rate ratio of suicide attempt among mothers when compared with the corresponding week among fathers in pre-conception and postpartum period, after excluding week 53

A restricted cubic spline with 4 knots, placed at 5th percentile, 35th percentile, 65th percentile and 95th percentile of distribution of weeks on the relevant x axis, was used to estimate the incidence rate ratios. The incidence rate ratios were adjusted for country of birth, age, calendar year, education level, civil status, category of income, primiparity, history of psychiatric disorders and history of suicide attempt, all derived at the start of each period. The shaded areas indicate 95% confidence interval.

We made changes in the manuscript accordingly, as described below.

Modified text:

Page 17: In addition, due to the small number of person-years, we excluded week 53 from analyses, and calculated SIR by week, and estimated IRRs comparing antepartum and

postpartum to preconception, and comparing mothers to fathers in preconception and postpartum.

Page 7: Furthermore, when excluding week 53 from the analysis, the rise in paternal absolute IR in late postpartum period was less pronounced yet the CIs overlapped with those from the primary analysis (Figure S11) whereas the IRRs changed minimally (Figure S12-S13).

<References: Does this manuscript reference previous literature appropriately? If not, what references should be included or excluded?>

Statistical literature is appropriate

<Clarity and context: Is the abstract clear, accessible? Are abstract, introduction and conclusions appropriate?>

Yes

My expertise is primarily in statistical methods rather than the subject matter, so I restricted my assessment to those areas.

No code was supplied

Editor's comments

- Please consider Reviewer #3's suggestion of using single estimates from the pre-conception period as default analysis and the current analyses with weekly pre-conception data as supplementary, provide full and appropriate justification for your chosen analytical approach, and present the results of the alternative analysis in your supplementary material.

Response: Thank you for the suggestion. We agree with the reviewer that using the average rate in preconception period as the reference group would help simplify the analytic approach. However, as mentioned by the reviewer, there is seasonal variation in suicidal behavior, which has been well documented in literature^[1-3]. Moreover, as illustrated in the current Table 1, there is seasonal variation in terms of the timing of conception/pregnancy in Sweden, likely due to the extensive summer vacation throughout the country. Therefore, the potential confounding effect from season should be controlled for in our analysis, for instance through the weekly comparison. However, if we used a single preconception estimate as the reference which averages the rate over a year/four seasons, we won't be able to control for season.

We have now adjusted for season in our analyses and updated related figures and supplementary files. Of note, in our analysis, we adjusted for season at the start of each period instead of season at each week; however, since the time from the start of the period to each week is consistent across all periods, this approach is virtually equivalent to adjustment for season at each week. The following changes have been made in manuscript:

Modified text:

Page 16: To compare the week-specific incidence rates of suicide attempt in the antepartum and postpartum periods with corresponding weeks in the preconception period, we used multivariable Poisson regression to estimate the incidence rate ratio (IRR) and 95% confidence interval (CI) of maternal and paternal suicide attempt. Comparing weekly rates allows us to control for season, which is a potential confounder to the studied association⁶². IRRs were adjusted for country of birth, age, calendar year, education level, civil status, category of income, primiparity, history of psychiatric disorders, history of suicide attempt and season, all derived at the start of each period.

Page 16-17: Similarly, to illustrate sex differences in the risk of suicide attempt in each period, we used multivariable Poisson regression to estimate IRR of suicide attempt, comparing mothers to fathers, weekly from preconception throughout postpartum period. To shed light on absolute sex differences, we further used Poisson regression to estimate the weekly incidence rate difference (IRD)⁶⁴. To visualize IRRs and IRDs, we also applied restricted cubic spline on week, and placed 4 knots at 5th percentile, 35th percentile, 65th percentile and 95th percentile of distribution of weeks on the relevant x axes. The parameterization can be found in eMethods in Supplementary files. 95% confidence interval was calculated based on standard error estimated from delta method and z value. The models were adjusted for country of birth, age, calendar year, education level, civil status, category of income, parity, history of psychiatric disorders, history of suicide attempt, and season, all derived at the start of each period.

References:

1. Ambar Akkaoui M, Chan-Chee C, Laaidi K, et al. Seasonal changes and decrease of suicides and suicide attempts in France over the last 10 years. *Sci Rep.* 2022;12:8231. doi:10.1038/s41598-022-12215-3
2. To S, Messias E, Burch L, Chibnall J. Seasonal variation in suicide: age group and summer effects in the United States (2015–2020). *BMC Psychiatry.* 2024;24:856. doi:10.1186/s12888-024-06309-7
3. Holopainen J, Helama S, Björkenstam C, Partonen T. Variation and seasonal patterns of suicide mortality in Finland and Sweden since the 1750s. *Environ Health Prev Med.* 2013;18(6):494-501. doi:10.1007/s12199-013-0348-4

- Reviewer #3 also raised concerns about the cubic splines analysis. Please add the full code you used for cubic splines to the github repository, explain in more depth about this analysis, in particular, respond to Reviewer #3's concern about the choice you made to use the same three knots for all three periods. Consider adding LOWESS smoothing there to be consistent.

Response: The point is well taken. The codes we used to generate splines were based on individual data and saved in separate files as main figures. Now we have integrated the codes and uploaded them to Github (<https://github.com/yihuiyang2/ParentalSuicide>). In addition, we applied the same knots to all periods. However, the results without using spline showed similar pattern as **Figure 3**, which illustrate that the use of spline should not affect our main conclusion. Nevertheless, we have now rerun the analysis for Figure 3, which used knots generated in each period separately. Not surprisingly, the pattern of IRR across week is similar to our previous Figure 3, and our main conclusion would therefore remain unchanged. We have also updated results in supplementary materials, including parameter results, Figure S7, FigureS8, Figure S12, and Table S3.

Figure 3 Incidence rate ratio of parental suicide attempt during and after pregnancy when compared with the corresponding week before pregnancy: a nationwide register-based study in Sweden, 2001-2021

The follow-up during pregnancy was censored at week 40 due to few subsequent cases. The dots represent the actual incidence rate ratios, and the error bars represent their 95% confidence interval. The dashed lines represent results when using restricted cubic spline with 4 knots, placed at 5th percentile, 35th percentile, 65th percentile and 95th percentile of distribution of weeks on the relevant x axes. The shaded areas indicate 95% confidence interval of the smoothed results. Note that a logarithmic scale was used. The incidence rate ratios were adjusted for country of birth, age, calendar year, education level, civil status, category of income, primiparity, history of psychiatric disorders, history of suicide attempt and season, all derived at the start of each period.

REVIEWER COMMENTS:

Reviewer #3 (Remarks to the Author):

Point a)

I think my misunderstanding was due to the omission of the word “half”. If I understand correctly, consider where imputation was required for a birth to a mother of a given parity and delivery method. Let us assume that the median length of stay for all births of the same parity and delivery method in the independent database was N days, then the best estimate of delivery date would be admission date plus N/2 (half way between admission and discharge). (If you added N to the admission date, you would get to the typical discharge date). It is evident that you have made the calculation as median/2 – it was just the omission of “half” in the text which caused confusion. There is no need for the addition of the text “., since the delivery date must come on or after the admission date, and we assumed that the delivery happened in the middle of the stay.”

Response: Thank you for the clarification. We have now removed the text.

Point b)

thank you for the clarification.

Point c)

Thank you for noting this. I now understand that you applied LOWESS to the 95% SIR confidence bounds. My query in e) was due to the misunderstanding that you had smoothed SIR by LOWESS, and then calculated the 95% bounds of the LOWESS.

Point d) Point e)

Thank you for these clarifications.

Point f)

Thank you for this change

Point g)

Thank you for standardizing S7 and S9. S2 appears to have different scales, with 1 at different heights above the x axis.

Response: Thank you for noticing this. We have now replotted Figure S2.

Figure S2 Incidence rate of parental suicide attempt before, during, and after pregnancy, by age at start of each period

Locally Weighted Scatterplot Smoothing (bandwidth=0.8) was used to smooth the lines. The shaded areas indicate 95% CI. Note that a logarithmic scale was used.

Point h)

Thank you. Please also correct this in the methods.

Response: Thank you for the suggestion. We have now clarified it in the Methods.

Modified text:

Page 16: To compare the week-specific incidence rates of suicide attempt in the antepartum and postpartum periods with corresponding weeks in the preconception period, we used multivariable Poisson regression to estimate the incidence rate ratio (IRR) and 95% confidence interval (CI) of maternal and paternal suicide attempt...To investigate a potential nonlinear relationship between IRRs and week, we applied restricted cubic spline on week, and placed 4 knots at 5th percentile, 35th percentile, 65th percentile and 95th percentile of distribution of weeks on the relevant x axes⁶³, to visualize weekly IRRs.

Page 16-17: Similarly, to illustrate sex differences in the risk of suicide attempt in each period, we used multivariable Poisson regression to estimate IRR of suicide attempt, comparing mothers to fathers, weekly from preconception throughout postpartum period... To visualize IRRs and IRDs, we also applied restricted cubic spline on week, and placed 4 knots at 5th percentile, 35th percentile, 65th percentile and 95th percentile of distribution of weeks on the relevant x axes.

Point i)

Thank you

Point j)

I remain unclear of the process used to incorporate cubic splines. The code for their

estimation is not included in the github repository. Thank you for including the parameters listed. However, they suggest the same knots were used for all three periods. Given the pregnancy period is only 40 weeks, compared to 52 weeks for the pre-conception and postpartum periods, this infers inconsistency.

Response: The point is well taken. The codes we used to generate splines were based on individual data and saved in separate files as main figures. Now we have integrated the codes and uploaded them to Github (<https://github.com/yihuiyang2/ParentalSuicide>). In addition, we applied the same knots to all periods. However, the results without using spline showed similar pattern as **Figure 3**, which illustrate that the use of spline should not affect our main conclusion. Nevertheless, we have now rerun the analysis for Figure 3, which used knots generated in each period separately. Not surprisingly, the pattern of IRR across week is similar to our previous Figure 3, and our main conclusion would therefore remain unchanged. We have also updated results in supplementary materials, including parameter results, Figure S7, FigureS8, Figure S12, and Table S3.

Figure 3 Incidence rate ratio of parental suicide attempt during and after pregnancy when compared with the corresponding week before pregnancy: a nationwide register-based study in Sweden, 2001-2021

The follow-up during pregnancy was censored at week 40 due to few subsequent cases. The dots represent the actual incidence rate ratios, and the error bars represent their 95% confidence interval. The dashed lines represent results when using restricted cubic spline with 4 knots, placed at 5th percentile, 35th percentile, 65th percentile and 95th percentile of distribution of weeks on the relevant x axes. The shaded areas indicate 95% confidence interval of the smoothed results. Note that a logarithmic scale was used. The incidence rate ratios were adjusted for country of birth, age, calendar year, education level, civil status, category of income, primiparity, history of psychiatric disorders, history of suicide attempt and season, all derived at the start of each period.

In addition, the 95% confidence bounds on related Figures (e.g. Figure 3 and figure 4, others in supplementary) appear quite narrow, considering the variability of the raw data (Figure 1 and Figure 2). Perhaps the standard errors of the smoothed spline

estimates have not been incorporated into the total variation after adjustment for country of birth, age, etc?

Response: Thank you for bringing this to our attention. First, we want to clarify that Figure 1-2 and Figure 3-4 are different outputs from different statistical approaches and therefore the CIs are not really comparable. Namely, Figure 1 and Figure 2 show the standardized incidence rate (SIR). The standard errors were calculated based on *Cochran, 1977*^[1], and represent how precise we estimated the absolute incidence rate. On the other hand, Figure 3 and Figure 4 plot incidence rate ratio (IRR) and incidence rate difference (IRD). The standard errors here were calculated according to delta method, and capture the statistical uncertainty of the relative difference. For instance, even we have a wide CI for incidence rate at some week in Figure 2, as long as the rate and CI of the mothers are clearly far away from that of the fathers, the IRR (relative manner) will land on a narrow CI in Figure 4. Therefore, the standard error, or 95% confidence interval in Figure 1-2 and Figure 3-4 is not comparable.

In addition, the procedure generating splines are simply data transformation and no standard error were generated. In addition, the model was fit with the spline variable and all other covariates as independent variables. Their coefficients and standard errors can be seen from the parameter results in the supplementary materials. Therefore, the standard error of the smooth splines should already be included in the total variation of the model.

Reference:

[1] Cochran WG. *Sampling Techniques*. 3rd ed. Wiley; 1977.

I understand the desire to smooth results to aid the reader in following broad patterns more easily, but I am not convinced that the given application of cubic splines is the best way to do so in this instance. As you show in Figure S14 and figure S15, the results without splines are consistent. There is more noise in the figure, but this is more realistic and less likely for the reader to infer more regularity in the pattern than we see. LOWESS smoothing could be repeated here

Response: Thank you for the suggestion. We agree with the reviewer that the results without splines are more realistic. However, we believe the smoothing results allow readers to quickly capture the pattern, which is important for clinicians to grasp the messages. Therefore, we have now presented both results in our main figures. Figure S14 and Figure S15 are removed.

Figure 3 Incidence rate ratio of parental suicide attempt during and after pregnancy when compared with the corresponding week before pregnancy: a nationwide register-based study in Sweden, 2001-2021

The follow-up during pregnancy was censored at week 40 due to few subsequent cases. The dots represent the actual incidence rate ratios, and the error bars represent their 95% confidence interval. The dashed lines represent results when using restricted cubic spline with 4 knots, placed at 5th percentile, 35th percentile, 65th percentile and 95th percentile of distribution of weeks on the relevant x axes. The shaded areas indicate 95% confidence interval of the smoothed results. Note that a logarithmic scale was used. The incidence rate ratios were adjusted for country of birth, age, calendar year, education level, civil status, category of income, primiparity, history of psychiatric disorders, history of suicide attempt and season, all derived at the start of each period.

Figure 4 Incidence rate ratio and incidence rate difference of suicide attempt among mothers when compared with the corresponding week among fathers: a nationwide register-based study in Sweden, 2001-2021

The follow-up during pregnancy was censored at week 40 due to few subsequent cases. The dots represent the actual incidence rate ratios and incidence rate differences, and the error bars represent their 95% confidence interval. The dashed lines represent results when using restricted cubic spline with 4 knots, placed at 5th percentile, 35th percentile, 65th percentile and 95th percentile of distribution of weeks on the relevant x axes. The shaded areas indicate 95% confidence interval of the smoothed results. The incidence rate ratios and incidence rate differences were adjusted for country of birth, age, calendar year, education level, civil status, category of income, primiparity, history of psychiatric disorders, history of suicide attempt and season, all derived at the start of each period.

Statistical methods, Point a)

I appreciate the authors comparing IRR rates in antepartum and postpartum to an average of the preconception period. In my view, this simplified approach is much more accessible to the reader. I would prefer this as the default analysis. I do not follow how comparing weekly rates would help to alleviate seasonal variation as the calculated rates for each week are an aggregation of hundreds of thousands of persons, presumably measured over the full range of seasons. If the authors are happy that the small significant declines in SIR for the preconception period (Table R1) are of negligible clinical concern, I strongly advocate that single measures of preconception suicide rate (one each for mothers and fathers) are used as reference measures when comparing weekly pregnancy rates and weekly postpartum rates.

In other words, either there is a decreasing trend in the preconception period, which justifies comparison of rates in each week of pregnancy to rates exactly 12 months before, OR there is no evidence of clinically relevant decline in preconception period, in which case it makes most sense to compare weekly pregnancy rates with a single preconception estimate.

Response: Thank you for the suggestion. We agree with the reviewer that using the average rate in preconception period as the reference group would help simplify the analytic approach. However, as mentioned by the reviewer, there is seasonal variation in suicidal

behavior, which has been well documented in literature^[1-3]. Moreover, as illustrated in the current Table 1, there is seasonal variation in terms of the timing of conception/pregnancy in Sweden, likely due to the extensive summer vacation throughout the country. Therefore, the potential confounding effect from season should be controlled for in our analysis, for instance through the weekly comparison. However, if we used a single preconception estimate as the reference which averages the rate over a year/four seasons, we won't be able to control for season.

We have now adjusted for season in our analyses and updated related figures and supplementary files. Of note, in our analysis, we adjusted for season at the start of each period instead of season at each week; however, since the time from the start of the period to each week is consistent across all periods, this approach is virtually equivalent to adjustment for season at each week. The following changes have been made in manuscript:

Modified text:

Page 16: To compare the week-specific incidence rates of suicide attempt in the antepartum and postpartum periods with corresponding weeks in the preconception period, we used multivariable Poisson regression to estimate the incidence rate ratio (IRR) and 95% confidence interval (CI) of maternal and paternal suicide attempt. Comparing weekly rates allows us to control for season, which is a potential confounder to the studied association. IRRs were adjusted for country of birth, age, calendar year, education level, civil status, category of income, primiparity, history of psychiatric disorders, history of suicide attempt and season, all derived at the start of each period.

Page 16-17: Similarly, to illustrate sex differences in the risk of suicide attempt in each period, we used multivariable Poisson regression to estimate IRR of suicide attempt, comparing mothers to fathers, weekly from preconception throughout postpartum period. To shed light on absolute sex differences, we further used Poisson regression to estimate the weekly incidence rate difference (IRD)⁶⁴. To visualize IRRs and IRDs, we also applied restricted cubic spline on week, and placed 4 knots at 5th percentile, 35th percentile, 65th percentile and 95th percentile of distribution of weeks on the relevant x axes. The parameterization can be found in eMethods in Supplementary files. 95% confidence interval was calculated based on standard error estimated from delta method and z value. The models were adjusted for country of birth, age, calendar year, education level, civil status, category of income, parity, history of psychiatric disorders, history of suicide attempt, and season, all derived at the start of each period.

References:

1. Ambar Akkaoui M, Chan-Chee C, Laaidi K, et al. Seasonal changes and decrease of suicides and suicide attempts in France over the last 10 years. *Sci Rep.* 2022;12:8231. doi:10.1038/s41598-022-12215-3
2. To S, Messias E, Burch L, Chibnall J. Seasonal variation in suicide: age group and summer effects in the United States (2015–2020). *BMC Psychiatry.* 2024;24:856. doi:10.1186/s12888-024-06309-7
3. Holopainen J, Helama S, Björkenstam C, Partonen T. Variation and seasonal patterns of suicide mortality in Finland and Sweden since the 1750s. *Environ Health Prev Med.* 2013;18(6):494-501. doi:10.1007/s12199-013-0348-4

Other responses

thank you for all other clarifications.

I confirm that I consider the overall end conclusions about the data to be sound. My personal preference is simplified methods (single pre-conception estimates, and minimal use of smoothing).

Response: Thank you. Please see our response above for our justification for the weekly comparison approach.

Reviewer #3 (Remarks on code availability):

the code is clear. however there is no code covering the calculation of cubic splines. the given code accesses the generated spline estimates from a data file, and does not appear to incorporate the standard errors of these estimates into the overall uncertainty of the model.

Response: We have now provided the codes to calculate smoothing splines. Regarding the question on standard error, please see our response at page 6.